# Combinatorial CRISPR screen reveals *FYN* and *KDM4* as targets for synergistic drug combination for treating triple negative breast cancer

Tackhoon Kim[1,2,3,4]*, Byung-Sun Park[1,2], Soobeen Heo[5], Heeju Jeon[1,2], Jaeyeal Kim[1,2], Donghwa Kim[6], Sang Kook Lee[6], So-Youn Jung[7], Sun-Young Kong[5,8], Timothy Lu[4]*

[1]Medicinal Materials Research Center, Korea Institute of Science and Technology, Seoul, Republic of Korea; [2]Department of Biological Sciences, Korea University, Seoul, Republic of Korea; [3]Division of Bio-Medical Science and Technology, Korea University of Science and Technology, Daejeon, Republic of Korea; [4]Research Lab of Electronics, Massachusetts Institute of Technology, Cambridge, United States; [5]Targeted Therapy Branch, Division of Rare and Refractory Cancer, Research Institute, National Cancer Center, Goyang, Republic of Korea; [6]College of Pharmacy, Natural Products Research Institute, Seoul National University, Seoul, Republic of Korea; [7]Division of Breast Surgery, Department of Surgery, National Cancer Center, Goyang, Republic of Korea; [8]Department of Laboratory Medicine, National Cancer Center, Goyang, Republic of Korea

*For correspondence:
tackhoon@kist.re.kr (TK);
tim@lugroup.org (TL)

Competing interest: The authors declare that no competing interests exist.

## eLife Assessment

This study presents a **valuable** finding that synthetically lethal kinase genes FYN and KDM4 may play a role in drug resistance to kinase inhibitors in TNBC. The evidence supporting the claims of the authors is **solid**, although the exploration of the upstream mechanisms regulating KDM4A or the downstream pathways through which FYN upregulation confers drug resistance would have strengthened the study. The work will be of interest to medical biologists working in the field of breast cancer.

**Abstract** Tyrosine kinases play a crucial role in cell proliferation and survival and are extensively investigated as targets for cancer treatment. However, the efficacy of most tyrosine kinase inhibitors (TKIs) in cancer therapy is limited due to resistance. In this study, we identify a synergistic combination therapy involving TKIs for the treatment of triple negative breast cancer. By employing pairwise tyrosine kinase knockout CRISPR screens, we identify *FYN* and *KDM4* as critical targets whose inhibition enhances the effectiveness of TKIs, such as NVP-ADW742 (IGF-1R inhibitor), gefitinib (EGFR inhibitor), and imatinib (ABL inhibitor) both in vitro and in vivo. Mechanistically, treatment with TKIs upregulates the transcription of *KDM4*, which in turn demethylates H3K9me3 at *FYN* enhancer for *FYN* transcription. This compensatory activation of *FYN* and *KDM4* contributes to the resistance against TKIs. *FYN* expression is associated with therapy resistance and persistence by demonstrating its upregulation in various experimental models of drug-tolerant persisters and residual disease following targeted therapy, chemotherapy, and radiotherapy. Collectively, our study provides novel targets and mechanistic insights that can guide the development of effective combinatorial targeted therapies, thus maximizing the therapeutic benefits of TKIs.

## Introduction

Tyrosine kinases have emerged as important drug targets in cancer therapy due to their druggability and pivotal roles in cell proliferation and survival (*Wu et al., 2016*). They are implicated in various aspects of cancer development (*Hanahan and Weinberg, 2011*), such as cell survival, proliferation, angiogenesis, and invasion, making them attractive targets for drug intervention. Consequently, tyrosine kinase inhibitors (TKIs) have gained considerable attention as primary agents for cancer treatment.

Triple negative breast cancer (TNBC) treatment has limited options for targeted therapy. TNBC, characterized by the absence of estrogen receptor, progesterone receptor, and HER2 expression, exhibits elevated activity of tyrosine kinases, including EGFR and IGF1R (*Li et al., 2021*; *Litzenburger et al., 2011*). However, several clinical trials investigating TKIs, such as VEGFR inhibitors, EGFR inhibitors, and FGFR inhibitors, in TNBC treatment have yielded disappointing results due to inadequate efficacy. Therefore, it is crucial to comprehend the mechanisms underlying TNBC's suboptimal response to TKIs to enable the development of more effective targeted therapies against TNBC.

The therapeutic efficacy of TKIs is compromised by intrinsic and acquired resistance (*Morgillo et al., 2016*). For instance, EGFR inhibitor gefitinib extended the median progression-free survival by only 5 months compared to conventional chemotherapy in non-small cell lung cancer (NSCLC) patients with EGFR mutation (*Maemondo et al., 2010*). Significant subset of drug resistance is driven by gene interactions that enable compensatory changes in signal transduction upon drug treatment. Compensatory activation of mitogenic signals, such as MET, PIK3CA amplification, and MAPK/ERK signaling activation, counterbalances the inhibition of EGFR by TKI osimertinib in a significant portion of NSCLC patients (*Leonetti et al., 2019*). Simultaneous inhibition of multiple signaling molecules that compensate for each other's loss is proposed as an effective strategy to overcome resistance to kinase inhibitor therapy, emphasizing the importance of combinatorial therapy (*Jin et al., 2023*).

Until recently, a highly scalable method for screening combinatorial therapy has been lacking. Combinatorial CRISPR screens have emerged as efficient tools to identify synergistic targets for combinatorial therapy. We and others recently developed combinatorial CRISPR screens to elucidate pairwise gene interactions (*Wong et al., 2016*; *Horlbeck et al., 2018*; *Han et al., 2017*; *Shen et al., 2017*). Our combinatorial genetic screen platform, combinatorial genetics *en Masse* (combiGEM), was successfully implemented to identify combinations of epigenetic regulators causing synthetic lethality in ovarian cancer cells (*Wong et al., 2016*).

In this study, we utilize CombiGEM-CRISPR technology to identify tyrosine kinase inhibitor combinations with synergistic effect in TNBC cell line and xenograft models for potential combinatorial therapy against TNBC. We highlight FYN as a key therapeutic target that, when inhibited, enhances the cytotoxic effect of inhibition of other tyrosine kinases (IGF1R, EGFR, and ABL2). Mechanistic studies reveal KDM4 as a crucial epigenetic regulator that demethylates H3K9me3 and transcriptionally upregulates *FYN* upon TKI treatment. In vitro and in vivo validation demonstrates the synergistic TNBC-shrinking effects of combining PP2, saracatinib (Src family kinase / FYN inhibitor) or QC6352 (KDM4 inhibitor) with TKIs. Additionally, we demonstrate the clinical significance of our findings by observing upregulation of *FYN* in various models of drug tolerant persisters and residual tumors after chemo-, radio-, or targeted therapy. Therefore, simultaneous targeting of *FYN-KDM4* and tyrosine kinase pathways through combinatorial therapy holds promise for effective therapy against TNBC.

## Results

### Pairwise tyrosine kinase knockout CRISPR screen reveals synergistic tyrosine kinase inhibition combination

For efficient translation of CRISPR screening data to drug combination, we selected 76 tyrosine kinases that could be inhibited by at least one drug from the drug repurposing hub database (*Supplementary file 1*; *Corsello et al., 2017*). For pairwise CombiGEM library construction, we chose three guide RNAs from the optimized Brunello single guide RNA (sgRNA) list (*Doench et al., 2016*) employing the iterative cloning method as previously described (*Wong et al., 2016*; *Wong et al., 2015*). The resulting library enabled screens of pairwise knockouts of the 76 tyrosine kinase genes, encompassing 54,289 sgRNA pairs representing 3003 pairwise gene disruptions (*Figure 1A*). To validate our library, we performed next-generation sequencing (NGS) and confirmed that 99.5% (2989/3003) of gene pairs

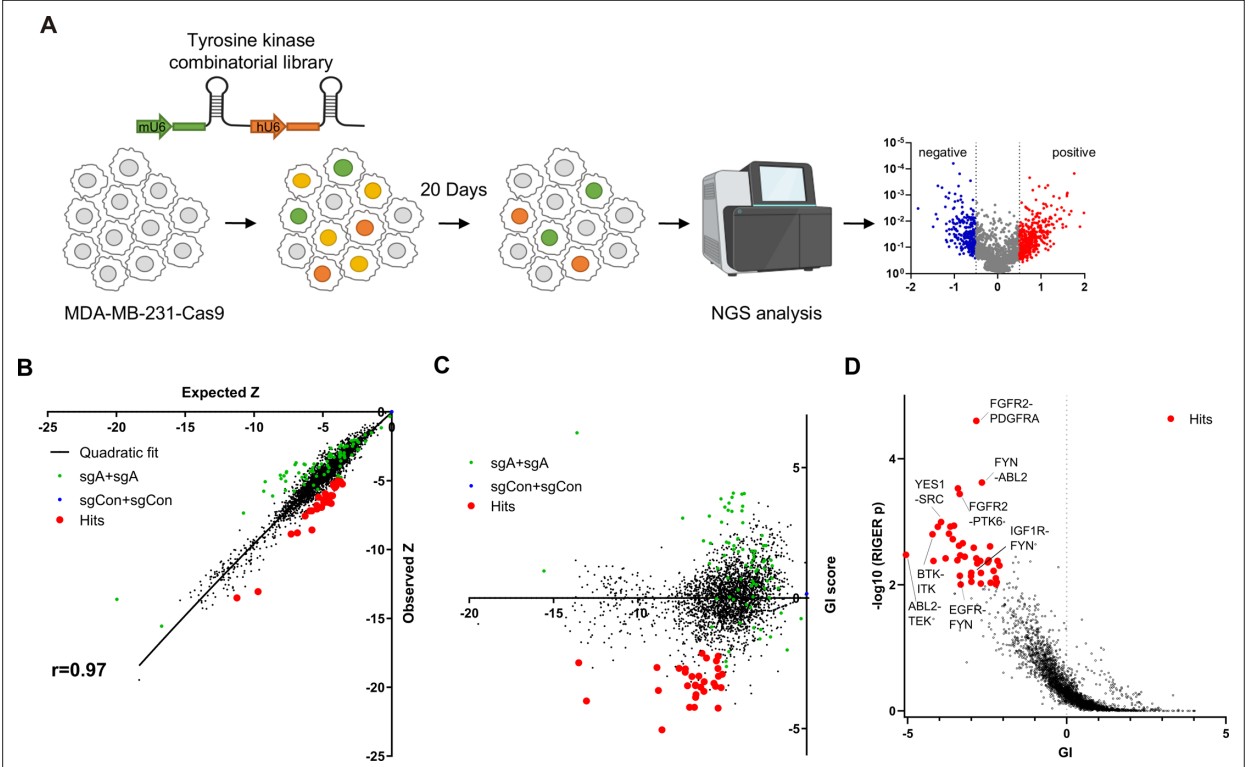

**Figure 1.** Pairwise CRISPR screen reveals combinations of synthetic lethal tyrosine kinase ablations. (**A**) Schematic diagram of combinatorial screens performed in TNBC cell line MDA-MB-231. (**B**) Scatter plot of expected growth phenotype Z score and observed growth phenotype Z score of each gene combination. Green dots indicate gene combinations where the identical gene is targeted by the two sgRNAs. Red dots indicate candidate synthetic lethal gene pairs listed in *Supplementary file 2*. (**C**) Scatter plot of growth phenotype Z score and normalized GI score of each gene combination. (**D**) Scatter plot of gene level GI score and RIGER p value calculated with GI scores of each sgRNA pairs that target the given gene pair.

The online version of this article includes the following figure supplement(s) for figure 1:

**Figure supplement 1.** Quality controls for combinatorial sgRNA libraries and screening results.

---

were represented by at least six pairs of sgRNAs, with the $\log_{10}$ reads per million of 0.5 (*Figure 1—figure supplement 1A*).

TNBC cell line MDA-MB-231 cells stably expressing Cas9 were transduced with the lentiviral library at low multiplicity of infection (MOI) of 0.3. Genomic DNA was harvested 3 days after transduction (designated as day 0[D0]), and 23 days after transduction (D20; *Figure 1A*) to perform PCR amplification of sgRNA pairs for subsequent NGS analysis. sgRNAs, instead of barcodes in our previous CombiGEM screens, were directly sequenced using paired-end sequencing to rule out uncoupling of barcodes of sgRNAs and barcodes (*Figure 1—figure supplement 1B and C*). We counted the occurrences of each sgRNA pair in the NGS data and calculated the normalized log2 fold change in counts between the day 20 and day 0 samples as the growth phenotype score Z (see Materials and methods). The Z scores for the two permutations of an sgRNA pair ($r=0.50$ between sgRNA-A +sgRNA B and sgRNA-B +sgRNA A pairs), the biological replicates, ($r=0.74$ between replicates #2 and #3), and independent sgRNA pairs targeting the same set of genes were positively correlated ($r=0.3–0.72$; *Figure 1—figure supplement 1D–G*).

Gene pairs that synergistically kill cells were identified by calculating gene interaction scores (GI). The GI scores were derived by comparing the growth phenotype score Z resulting from the disruption of a gene pair ($Z_{A+B}$, observed Z score) to the sum of Z scores obtained from the disruption of each gene individually within the pair ($Z_{A+Con} + Z_{B+Con}$, expected Z score; *Figure 1B*). The expected and observed Z scores for each gene (or sgRNA) pair exhibited a strong positive correlation ($r=0.97$ gene level, $r=0.88$ sgRNA level), suggesting that most random pairwise combinations of tyrosine kinase perturbations show additive effects (*Figure 1B*, *Figure 1—figure supplement 1H*). GI scores were calculated by quantifying each gene pair's normalized deviation from the quadratic fit of the expected-observed

Z score plot (*Horlbeck et al., 2018*; *Figure 1C*, *Figure 1—figure supplement 1I*, see Materials and methods).

We selected 30 synthetic lethal gene pairs using cutoffs for gene level GI score <-2 and p<0.01 for GI scores determined by RIGER analysis (*Luo et al., 2008*) and Z<-5 (*Figure 1D* and *Supplementary file 2*). Among these, the SRC-YES pair was one of the strongest synthetic lethal gene pairings (GI = –3.95). Notably, SRC-YES belong to the same tyrosine kinase family and are known to be functionally redundant and are expected to be synthetic lethal (*Stein et al., 1994*). These findings provide evidence for the effectiveness of our screening approach in identifying synthetic lethal gene pairs.

Synthetic lethal gene pairs are next validated by expressing the pair of sgRNAs targeting them. To achieve this, we introduced lentiviral vectors carrying two distinct sgRNAs targeting the candidate synthetic lethal pairs, each tagged with a different fluorescent protein (GFP and mCherry). MDA-MB-231 Cas9 cells were transduced with the lentivirus at a low titer (MOI ~0.5), resulting in a mixed population of cells expressing either one or both sgRNAs along with their respective fluorescent proteins (*Figure 2A*, *Figure 2—figure supplement 1A*). We monitored the decrease in the number of GFP/mCherry double-positive cells to evludate synthetic lethality (see Materials and methods). We validated the efficacy of the sgRNAs used in *Figure 2A* through the T7 endonuclease assay, which confirmed efficient gene editing (*Figure 2—figure supplement 1B*). Consistent with our CRISPR screening results, we observed that the disruption of six out of eight synergistic target gene combinations led to a reduction in cell viability beyond what was predicted by the Bliss independence model (*Figure 2B*). Moreover, the relative viability of double knockout cells and the rate of synergistic killing demonstrated a strong correlation with our screening data (*r*=0.65 for both viability and synergistic effect; *Figure 2—figure supplement 1C*). Collectively, our findings provide compelling evidence that our screening approach successfully identified synthetic lethal gene pairs with a high level of confidence.

## FYN inhibition synergizes with IGF1R, EGFR, ABL2 inhibitions in cell killing

We noticed that several validated synergistic target gene pairs included *FYN* (e.g. *FYN* +IGF1 R, *FYN* +EGFR, and *FYN* +ABL2). Notably, network analysis of the 30 candidate synergistic tyrosine kinase pairs revealed that *FYN* is one of the key nodes participating in synergistic interactions with multiple genes (*Figure 2C*). Expression of FYN, a member of Src family kinase, has been implicated in cancer malignancy including drug resistance (*Irwin et al., 2015*; *Fenouille et al., 2010*; *Airiau et al., 2017*; *Grosso et al., 2009*). Particularly, recent studies highlighted significant contribution of FYN in TNBC malignancy by promoting epithelial-to-mesenchymal transition (EMT; *Lee et al., 2018*; *Xie et al., 2016*). Interestingly, we found that *FYN*, but not *SRC*, exhibited significant upregulation in TNBC compared to other subtypes, as evidenced by microarray data from primary tumor samples (*Hatzis et al., 2011*) and the cancer cell line encyclopedia (CCLE; *Ghandi et al., 2019*; *Figure 2D and E*). In contrast, other key nodes in *Figure 2C*, including FGFR2, FRK, and TEK were not expressed at appreciable levels in MDA-MB-231 (log2(TPM + 1) for TEK: 0.0704, FRK:0.124, FGFR2:0.227), and their expressions were not significantly upregulated in TNBCs compared to other breast cancer subtypes (*Figure 2—figure supplement 2*). Therefore, we proceeded further in focusing on validating FYN as key candidate synthetic lethal gene. These findings suggest that FYN could represent an attractive drug target for TNBC treatment. To investigate this further, we assessed whether simultaneous inhibition of FYN by PP2, which selectively targets the SRC family kinase inhibitor with the highest potency against FYN, in combination with other kinase inhibitors (TKIs), could inhibit cancer cell growth (*Hanke et al., 1996*). PP2 as a single agent significantly downregulated MDA-MB-231 cell viability (*Figure 2—figure supplement 3A*). Therefore, we focused on synergistic cell death by TKI combinations above additive effects by each TKI. To this end, analysis using Synergy-Finder plus (*Zheng et al., 2022*) revealed that all TKI combinations involving PP2 and NVP-ADW742 (IGF1R inhibitor), gefitinib (EGFR inhibitor) or imatinib (ABL inhibitor) synergistically induced cell death in MDA-MB-231 cells (*Figure 2F*). Dose-response curves demonstrated that co-treatment with PP2 reduced the IC50 of the tested TKIs by 34–61%, indicating that PP2 sensitized cancer cells to TKI treatment (*Figure 2G*). Similar synergy was observed when TKI combinations included saracatinib, an SRC family kinase inhibitor (*Green et al., 2009*), in place of PP2 (*Figure 2—figure supplement 3B*). Moreover, specific ablation of *FYN*, but not *SRC*, sensitized cells to TKIs, highlighting the critical role

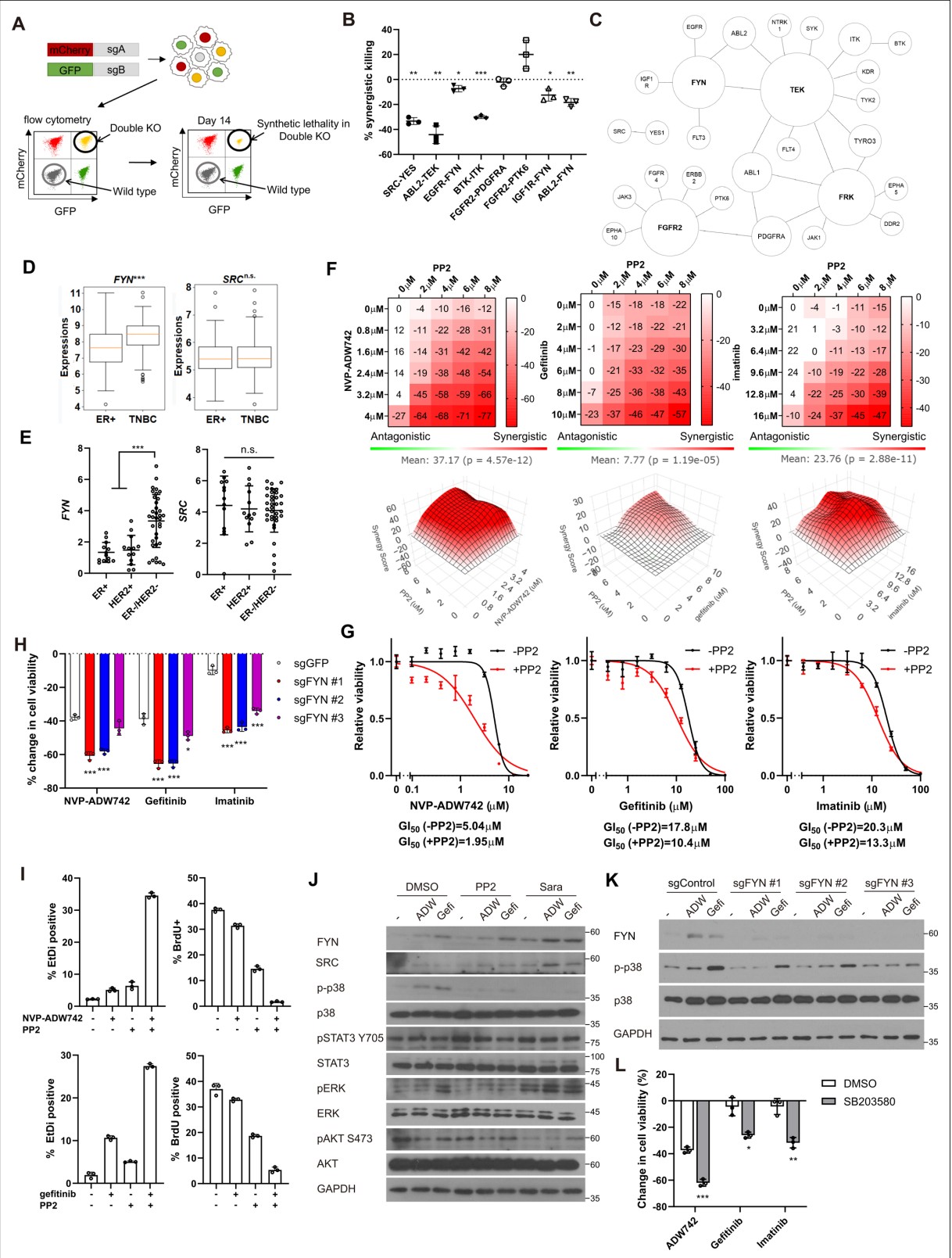

**Figure 2.** FYN is critical mediator of TKI resistance. (**A**) Schematic diagram of in vitro validation of synthetic lethal gene pairs using sgRNAs. (**B**) Summary of synergistic killing by sgRNAs targeting indicated gene pairs (n=3). (**C**) Network analysis of the 30 candidate synthetic lethal gene pairs highlighted in *Figure 1D*. The size of each node is manually drawn to be proportional to the number of connections the gene has. (**D, E**) *FYN* and SRC mRNA expressions in (**D**) microarray data of primary breast cancers of GSE25066 cohort, and (**E**) in cancer cell line encyclopedia, for indicated subtypes.

*Figure 2 continued on next page*

Figure 2 continued

(**F**) Summary of MTT assay with MDA-MB-231 cells treated with the TKI combinations at indicated concentrations (n=2). Synergistic killing is calculated using SynergyFinder with Bliss independence model. Raw data for the analysis is available in *Figure 2—source data 1*. (**G**) Dose response curve of the indicated TKI in the presence and absence of 5 µM PP2 treated for 72 hr (n=3). (**H**) MTT assay with MDA-MB-231 Cas9 cells expressing indicated sgRNAs treated with indicated TKIs for 72 hr (n=3). (**I**) Cell death and cell proliferation in MDA-MB-231 cells treated with NVP-ADW742, gefitinib and PP2 either as single agent or as combination for 72 hr (n=3). (**J**) Western blot analysis of MDA-MB-231 cells treated with indicated drugs for 48 hr. (**K**) Western blot analysis of MDA-MB-231 Cas9 cells expressing indicated sgRNA and treated with indicated drugs for 48 hr. (**L**) MTT assay of MDA-MB-231 cells treated with indicated drugs for 72 hr (SB203580: 10 µM, NVP-ADW742: 4 µM, gefitinib: 10 µM, imatinib: 10 µM) (n=3). PP2, Saracatinib, NVP-ADW742, gefitinib, and imatinib were treated at 5 µM, 5 µM, 4 µM, 12 µM, 12 µM unless otherwise indicated. All data are plotted as mean ± s.d. One sample t-test for B, and unpaired two-sided Student's t-test in D,E,H, and L. *, $p<0.05$; **, $p<0.01$; ***, $p<0.001$; n.s., $p>0.05$. All replicates are biological replicates.

The online version of this article includes the following source data and figure supplement(s) for figure 2:

**Source data 1.** Original files for western blot analysis displayed in *Figure 2J and K*.

**Source data 2.** Image files containing original western blots for *Figure 2J and K*, indicating the relevant bands.

**Figure supplement 1.** Validation of screening hits.

**Figure supplement 1—source data 1.** Original files for gel image displayed in *Figure 2—figure supplement 1B*.

**Figure supplement 1—source data 2.** Image files for gel image displayed in *Figure 2—figure supplement 1B*, indicating the relevant bands.

**Figure supplement 2.** Expression of TEK, FRK, and FGFR2 in breast cancer cell lines of different subtypes.

**Figure supplement 3.** FYN inhibition synergizes with TKIs in multiple TNBC cell lines.

**Figure supplement 3—source data 1.** Original files for western blot analysis displayed in *Figure 2—figure supplement 3I and J*.

**Figure supplement 3—source data 2.** Image files containing original western blot analysis displayed in *Figure 2—figure supplement 3I and J*, indicating the relevant bands.

of FYN as a member of SRC kinase family responsible for TKI resistance (*Figure 2H*, *Figure 2—figure supplement 3C*). Ablation of FYN itself did not significantly decrease cell viability (*Figure 2—figure supplement 3D*). Importantly, we observed similar synergy between the same drug combinations in other TNBC cell lines, including Hs578T, HCC1143, HCC1395, and HCC1937 cells (*Figure 2—figure supplement 3E–H*). Further assessment using live-dead and BrdU assays revealed that both the PP2 + NVP-ADW742 and PP2 +gefitinib combinations synergistically induced cell death while inhibiting cell growth (*Figure 2I*).

Persistent activation of MAPK pathway and PI3K-AKT pathway has been associated with TKI resistance in various cancers (*Morgillo et al., 2016*). Therefore, we investigated which downstream pathways were involved in sensitizing cells to TKI treatment. Notably, the p38 MAPK was significantly attenuated following treatment with either PP2 or saracatinib treatment (*Figure 2J*). Previous studies with imatinib resistant CML cells identified ERK signaling as critical downstream of FYN activation (*Fenouille et al., 2010*; *Airiau et al., 2017*). However, FYN inhibition failed to significantly downregulate phosphorylated ERK level upon imatinib treatment, indicating downstream signals of FYN leading to drug resistance may be context dependent. Genetic ablation of *FYN* similarly reduced p38 activation (*Figure 2K*). Attenuation of p38 activity was also observed in an independent TNBC cell line, Hs578T (*Figure 2—figure supplement 3I and J*). Importantly, treatment of p38 MAPK pathway inhibitor SB203580 markedly sensitized cells to TKI treatment (*Figure 2L*), while SB203580 as single agent did not significantly change cell viability (*Figure 2—figure supplement 3K*).

## FYN mRNA is induced upon TKI treatment in KDM4 dependent manner

Our discovery that inhibition of FYN synergizes with multiple TKIs possessing distinct target profiles suggests that FYN may play a role in general resistance mechanisms against TKI therapy. Consistently, we observed an increase in both protein and mRNA levels of *FYN* following TKI treatment, indicating that upregulation of FYN confers compensatory survival signal in TKI-treated cells (*Figure 3A–B*). The phosphorylation levels of FYN was increased proportional to FYN protein level, indicating specific kinase activity of FYN did not change (*Figure 3—figure supplement 1A*). Previous study suggested that increased expression of EGR1 transcription factor is responsible for FYN mRNA accumulation in imatinib-resistant CML (*Irwin et al., 2015*). Consistently, EGR1 expression was increased upon TKI treatment in MDA-MB-231 cells. However, EGR1 expression was not increased in TKI treated MDA-MB-231 cells, nor did its knockout significantly downregulated FYN mRNA levels (*Figure 3—figure supplement 1B*). To elucidate the mechanisms underlying the accumulation of *FYN*, we treated

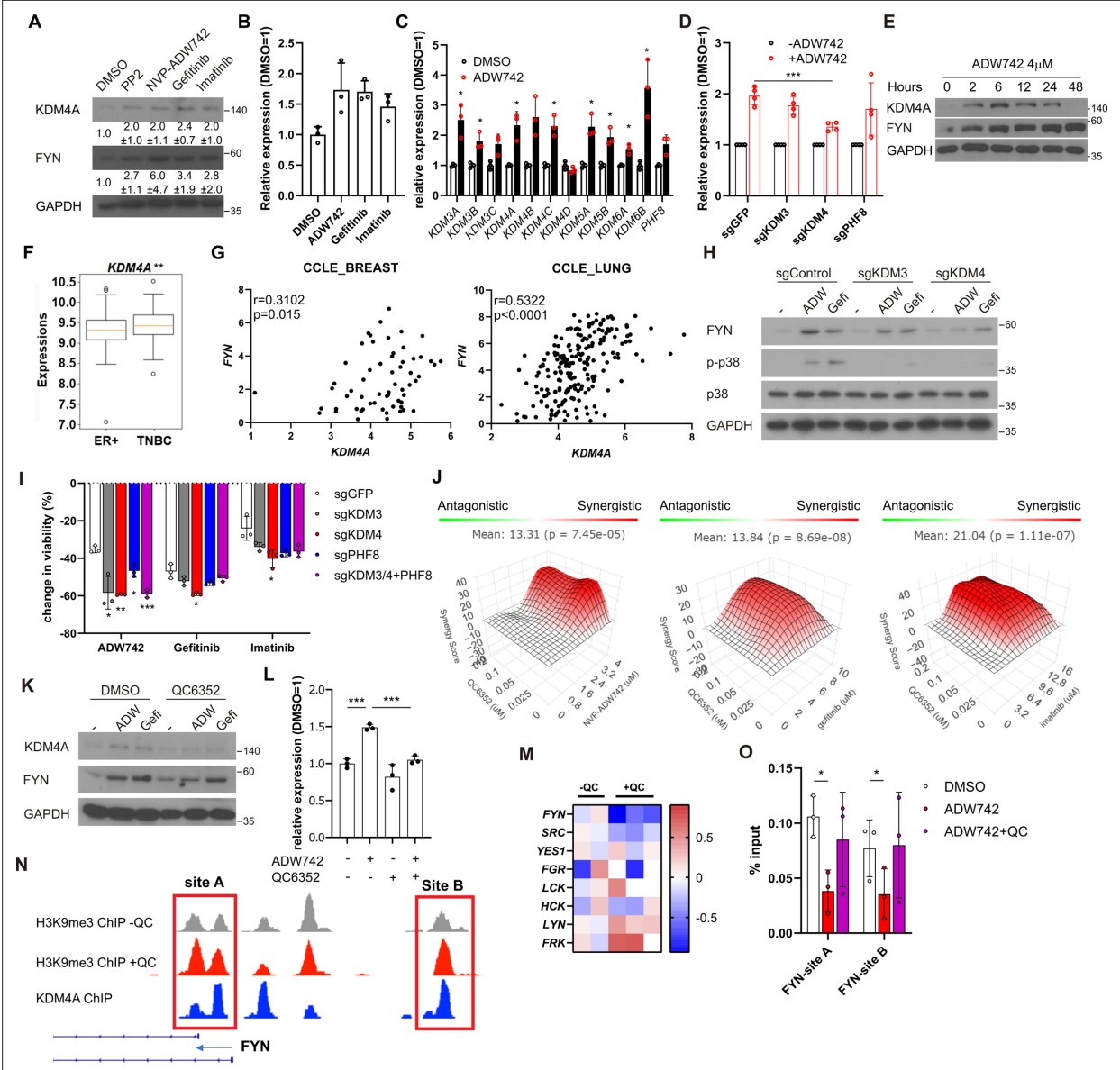

**Figure 3.** Activation of *KDM4* upregulates *FYN*, conferring drug resistance. (**A**) Western blot analysis of MDA-MB-231 cells treated with indicated drugs. Numbers below blots indicate quantification of average ± s.d. expression level normalized to GAPDH in three independent experiments. (**B**) RT-qPCR analysis of *FYN* expression levels in MDA-MB-231 cells treated with indicated drugs for 48 hr (n=3). (**C**) RT-qPCR analysis of indicated jumonji family histone demethylase expression levels after 48 hr treatment of NVP-ADW742 (n=3). (**D**) Changes in *FYN* mRNA levels upon 48 hr of NVP-ADW742 treatment in MDA-MB-231 Cas9 cells expressing indicated sgRNAs (n=4). (**E**) Western blot analysis of MDA-MB-231 cells treated with NVP-ADW742 for indicated time. (**F**) *KDM4A* mRNA levels in primary tumor tissues of indicated subtypes in GSE25066 cohort. (**G**) Positive correlation of *FYN* and *KDM4A* mRNA levels in CCLE database. (**H**) Western blot analysis of MDA-MB-231 Cas9 cells expressing indicated sgRNA and treated with indicated drugs for 48 hr. (**I**) MTT assay of MDA-MB-231 Cas9 cells expressing indicated sgRNAs and treated with indicated drugs for 72 hr (n=3). (**J**) SynergyFinder analysis of MDA-MB-231 cells treated with indicated drug combinations (n=2). (**K, L**) Western blot analysis (**K**) and RT-qPCR analysis (**L**) of MDA-MB-231 cells treated with indicated drugs for 48 hr (n=3). (**M**) mRNA expression levels of SRC family kinases in breast cancer stem cells treated with QC6352 in RNA sequencing data described in *Metzger et al., 2017* (**N**) H3K9me3 and KDM4A enrichment at genomic locus encoding *FYN* promoter in ChIP sequencing data described in the same study as (**M**). (**O**) H3K9me3 Chromatin immunoprecipitation-qPCR analysis of MDA-MB-231 cells treated with indicated drug for 48 hr at specified genomic loci (n=3). QC6352, PP2, NVP-ADW742, gefitinib and imatinib were treated at 10 μM, 5 μM, 4 μM, 12 μM, 12 μM, respectively, unless otherwise indicated. All data are plotted as mean ± s.d. Unpaired two-sided Student's t-test in B, C, D, F, I, and L. Paired two-sided Student's t-test in O. *, p<0.05; **, p<0.01; ***, p<0.001; n.s., p>0.05. All replicates are biological replicates.

The online version of this article includes the following source data and figure supplement(s) for figure 3:

**Source data 1.** Original files for western blot analysis displayed in *Figure 3A, E, H and K*.

*Figure 3 continued on next page*

*Figure 3 continued*

**Source data 2.** Image files containing original western blots for *Figure 3A, E, H and K*, indicating the relevant bands.

**Figure supplement 1.** KDM4 regulates FYN expression level, sensitizing MDA-MB-231 cells to TKIs.

**Figure supplement 1—source data 1.** Original files for western blot analysis displayed in *Figure 3—figure supplement 1A, B and F*.

**Figure supplement 1—source data 2.** Image files containing original western blot analysis displayed in *Figure 3—figure supplement 1A, B and F*, indicating the relevant bands.

**Figure supplement 2.** KDM4 inhibition synergizes with TKIs in multiple TNBC cell lines.

**Figure supplement 2—source data 1.** Original files for gel image displayed in *Figure 3—figure supplement 2B*, and western blot analysis displayed in *Figure 3—figure supplement 2C and F*.

**Figure supplement 2—source data 2.** Images files for gel image displayed in *Figure 3—figure supplement 2B*, and western blot analysis displayed in *Figure 3—figure supplement 2C and F*, indicating the relevant bands.

MDA-MB-231 cells with inhibitors targeting key epigenetic modifiers and assessed their synergistic effects with NVP-ADW742 in cell killing, as well as their impact on *FYN* mRNA accumulation. Multiple drugs, including pinometostat (DOT1L inhibitor *Daigle et al., 2013*), tazemetostat (EZH2 inhibitor *Knutson et al., 2013*), A366 (G9a inhibitor *Sweis et al., 2014*) and GSK-J4 (KDM6 inhibitor *Kruidenier et al., 2012*) strongly decreased cell viability upon TKI treatment (*Figure 3—figure supplement 1C*). As the increase in *FYN* mRNA is responsible for TKI resistance, we reasoned that the drug that directly affect FYN mRNA level and hence cell viability should be an inhibitor of epigenetic regulator that enhances transcription. To this end, we focused on pinometostat and GSK-J4 for further validations. Intriguingly, while GSK-J4 decreased FYN mRNA upon NVP-ADW742 treatment, pinometostat failed to decrease it (*Figure 3—figure supplement 1D*). Consistent with this, treatment with NVP-ADW742 increased the expression of most members of the jumonji domain histone demethylase family (*Figure 3C*). This observation is consistent with a previous study on taxane-resistant H1299 lung cancer cells (*Dalvi et al., 2017*), suggesting that histone demethylases may play critical roles in activating a drug resistance gene program. However, the ablation of KDM6, the primary targets of GSK-J4, failed to significantly decrease *FYN* expression (*Figure 3—figure supplement 1E*). GSK-J4 is known to inhibit other jumonji domain histone demethylase family proteins including KDM4 and KDM5 (*Heinemann et al., 2014*). Therefore, we tested the possibility that other histone demethylase may be involved in regulating *FYN* expression. Among jumonji domain histone demethylases, *KDM4*, and to a lesser extent *KDM3*, were the only gene family members whose ablation inhibited *FYN* upregulation and p38 activation upon TKI treatment (*Figure 3D*, *Figure 3—figure supplement 1F*). Ablation of KDM5, which has been shown to induce drug tolerance in cancer cells (*Sharma et al., 2010*), did not significantly alter *FYN* expression (*Figure 3—figure supplement 1G*). Like NVP-ADW742 treatment, gefitinib treatment increased *KDM4* demethylase levels (*Figure 3—figure supplement 1H*). We also analyzed two independent TNBC organoids obtained from primary tumors and found concurrent upregulation of *KDM4* with *FYN* mRNAs upon NVP-ADW742 and gefitinib treatment (*Figure 3—figure supplement 1I*). Critically, time course experiment with NVP-ADW742 treated MDA-MB-231 revealed that accumulation of KDM4A protein preceded FYN protein (*Figure 3E*), suggesting that KDM4A accumulation may be responsible for FYN accumulation. Both KDM3 and KDM4 demethylates methylated H3K9, thereby promoting the opening heterochromatin for transcription (*Hyun et al., 2017*). Remarkably, expression of *KDM4A*, the most abundantly expressed gene among KDM4 demethylases in TNBC cell lines (*Figure 3—figure supplement 2A*) was enriched in TNBC compared to other breast cancer subtypes (*Figure 3F*) and was positively correlated with *FYN* expression in CCLE database, suggesting that KDM4 regulates *FYN* mRNA levels (*Figure 3G*). Genetic ablation of *KDM3* or *KDM4* (*Figure 3—figure supplement 2B and C*) decreased *FYN* and p38 activity. Also, genetic ablation of KDM3 or KDM4 sensitizing MDA-MB-231 cells to TKIs (*Figure 3H–I*). The Ablation of KDM3 or KDM4 only had modest but statistically insignificant effect on cell viability (*Figure 3—figure supplement 2D*). Likewise, treatment of KDM4 inhibitor QC6352 (*Chen et al., 2017*) synergized with TKIs in killing MDA-MB-231 cells (*Figure 3J*). QC6352 treatment also significantly attenuated *FYN* accumulation upon NVP-ADW742 treatment (*Figure 3K and L*). This was consistent with the RNA sequencing data results in the previous study with breast cancer stem cells treated with QC6352 (*Metzger et al., 2017*). Specifically, *FYN* was the most significantly downregulated SRC family kinase upon QC6352 treatment (*Figure 3M*). Analysis of chromatin IP (ChIP) sequencing data from the same study revealed KDM4A

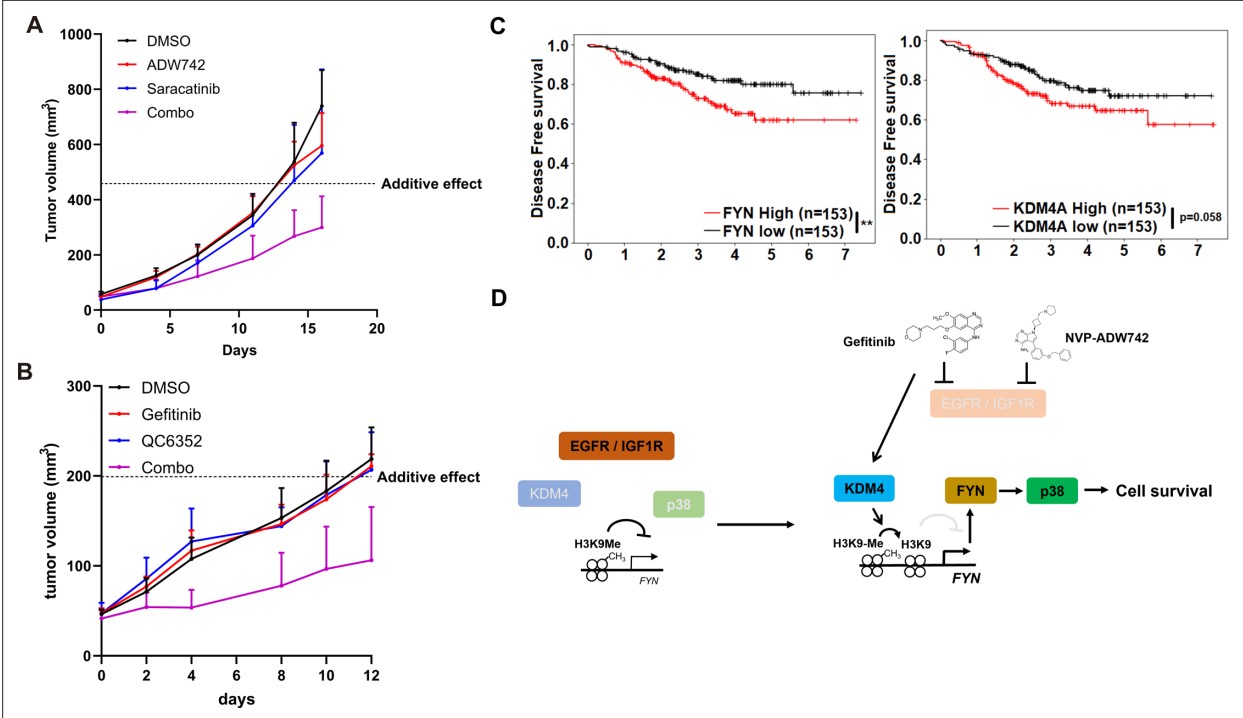

**Figure 4.** Combination therapy targeting FYN +IGF1 R and KDM4 +EGFR synergistically eliminates tumor in vivo. (**A, B**) Tumor volume for MDA-MB-231 xenografts treated with indicated drugs. Additive effects were calculated by Bliss independence model (n=5). (**C**) Distant relapse free survival of GSE25066 patient cohort classified by *FYN* (left) and *KDM4A* (right) mRNA expression. (**D**) Schematics diagram of the mechanism of KDM4-FYN conferring TKI resistance. All data are plotted as mean ± s.d. *, p<0.05; **, p<0.01; ***, p<0.001; n.s., p>0.05. All replicates are biological replicates.

The online version of this article includes the following figure supplement(s) for figure 4:

**Figure supplement 1.** No overt toxicity with drug combinations in xenograft experiments.

enrichment near FYN promoter; and QC6352 treatment increased H3K9me3 enrichment at the same locus (*Figure 3N*). Indeed, this FYN promoter locus exhibited a reduction in H3K9me3 following NVP-ADW742 treatment, while QC6352 treatment restored H3K9me3 enrichment (*Figure 3O*). This finding suggests that KDM4 may directly demethylate H3K9me3 at FYN promoter to upregulate *FYN* transcription. In contrast, H3K27me3 marks, which is demethylated by KDM6 family demethylases, were not significantly changed at FYN promoter upon NVP-ADW742 treatment (*Figure 3—figure supplement 2E*). *FYN* accumulation and resistance to TKIs were also confirmed to be attenuated by QC6352 treatment in other independent TNBC cell lines (*Figure 3—figure supplement 2F and G*).

## FYN/KDM4 inhibition synergizes with TKI treatment in vivo

We proceeded to investigate the potential clinical application of our synthetic lethal gene pairs as combinatorial therapy by assessing the in vivo efficacy of pharmacological interventions targeting these gene pairs using MDA-MB-231 xenograft models. Strikingly, co-treatment of saracatinib and NVP-ADW742 synergistically reduced tumor size, whereas treatment with either agent alone was ineffective in slowing tumor growth (*Figure 4A*). All treatment groups exhibited minimal changes in body weight, indicating that the overall health of the animals was not adversely affected by the combination treatment (*Figure 4—figure supplement 1A*). Saracatinib-gefitinib combination was not tested as saracatinib can inhibit EGFR (*Green et al., 2009*). Similarly, KDM4 inhibitor QC6352 synergized with gefitinib in reducing MDA-MB-231 xenograft tumor growth without causing overt changes in animal health (*Figure 4B*, *Figure 4—figure supplement 1B*). Additionally, the expression levels of *FYN* and *KDM4A* were found to be correlated with poor prognosis in a previously reported breast cancer cohort (*Hatzis et al., 2011*), highlighting the potential of targeting these two genes as therapeutic targets for TNBC (*Figure 4C*). Collectively, our results demonstrate that upregulation of *KDM4* upon TKI treatment reduces H3K9me3 mark in *FYN* enhancer, thereby increasing *FYN* expression and promoting cell survival under TKI treatment (*Figure 4D*).

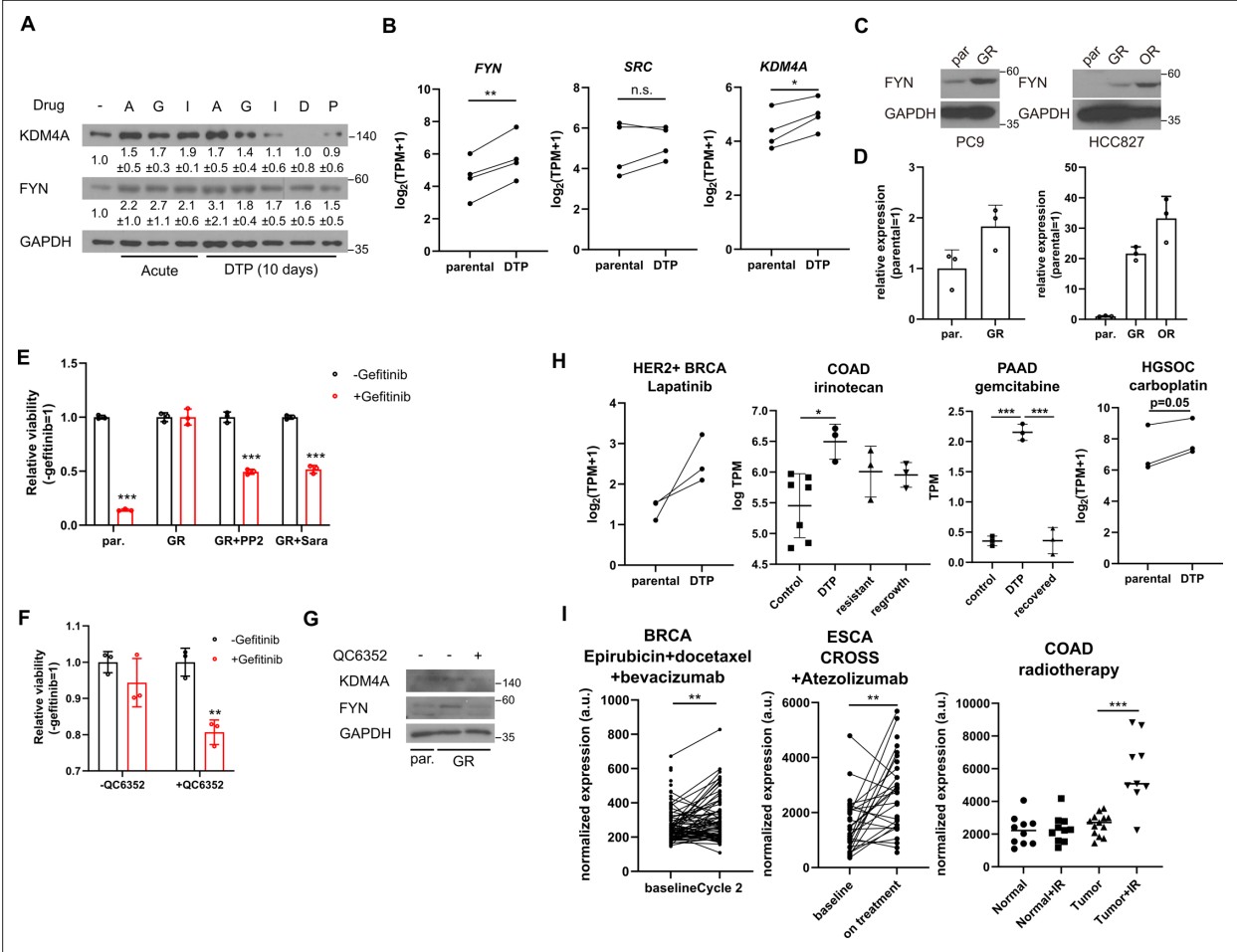

**Figure 5.** *FYN* and *KDM4* are associated with drug tolerance. (**A**) MDA-MB-231 cells treated with indicated drugs for short (acute: 2 days) and long (DTP: 10 days) time periods. A: 1 µM NVP-ADW742, G: 5 µM gefitinib, I: 5 µM imatinib, D: 100 nM doxorubicin, P: 5 nM paclitaxel. Numbers below blots indicate quantification of average ± s.d. expression level normalized to GAPDH in three independent experiments. (**B**) Summary of mRNA expressions of indicated genes in EGFR mutant lung cancer cells (parental) and their derivative osimertinib tolerant persisters (DTP). (**C, D**) Western blots (**C**) and RT-qPCR (**D**) analyses of indicated parental and EGFR inhibitor resistant lung cancer cells. (**E, F**) MTT assay with PC9 parental (par.) and gefitinib-resistant (GR) cells treated with indicated drug combinations (gefi: 2 µM gefitinib, PP2: 5 µM PP2, Sara: 5 µM saracatinib, QC6352: 10 µM QC6352) for 72 hr. (**G**) Western blot analysis with PC9 cells treated with 10 µM QC6352 for 48 hr. (**H**) *FYN* mRNA expression levels of parental and DTP populations in various cancers treated with indicated drugs (**I**) *FYN* mRNA expression levels of residual disease after indicated treatments. All data are plotted as mean ± s.d. Paired two-sided Student's t-test in B, H (HER2 +BRCA set and HGSOC carboplatin set), and I (BRCA set and ESCA set), and unpaired two-sided Student's t-test in H (COAD and PAAD sets) and I (COAD set). The expression data in B, H, and I are obtained from NCBI gene expression omnibus. The accession numbers of the expression data analyzed are listed in *Supplementary file 4*. *, p<0.05; **, p<0.01; ***, p<0.001; n.s., p>0.05. All replicates are biological replicates.

The online version of this article includes the following source data and figure supplement(s) for figure 5:

**Source data 1.** Original files for western blot analysis displayed in *Figure 5A, C and G*.

**Source data 2.** Image files containing original western blots for *Figure 5A, C and G*, indicating the relevant bands.

**Figure supplement 1.** KDM4 expression levels in drug tolerant cancers and residual tumor after therapy.

## FYN is associated with drug tolerant persister phenotype

The observed epigenetic alterations in regulators conferring resistance to multiple cancer drugs closely resemble non-genetic changes associated with the generation of drug-tolerant persisters (*Sharma et al., 2010*). Indeed, prolonged incubation of MDA-MB-231 cells treated with TKIs or conventional chemotherapy drugs such as doxorubicin or paclitaxel resulted in increased levels of *FYN* (*Figure 5A*). Curiously, *KDM4A* expression was only upregulated upon treatment with NVP-ADW742 and gefitinib, suggesting that while *FYN* upregulation is a general feature of drug tolerant

cells, the mechanism of *FYN* upregulation may vary depending on the specific drug being used. Analysis of previously published RNA sequencing data from a series of osimertinib tolerant EGFR mutated lung cancer cell lines (*Gogleva et al., 2022*) revealed higher expression levels of *FYN* and *KDM4A* in the drug persisters, but not SRC (*Figure 5B*). Consistently, we confirmed upregulation of *FYN* at both the protein and mRNA levels in gefitinib and osimertinib resistant PC9 and HCC827 cells (*Figure 5C and D*). Pharmacological inhibition of FYN or downregulation of *FYN* expression through inhibition of KDM4 sensitized gefitinib resistant PC9 cells to EGFR inhibitor, suggesting that *FYN-KDM4* are responsible for gefitinib resistant phenotype in this cell line (*Figure 5E–G*).

Importantly, upregulation of *FYN* has been consistently observed in multiple independent studies involving drug-tolerant cancer cell lines and patient-derived xenografts treated with various drugs that have distinct target profiles, including TKIs (lapatinib, a HER2 inhibitor, against HER2 positive breast cancer *Chang et al., 2022*) and chemotherapy drugs (irinotecan, topoisomerase inhibitor against colorectal cancer *Rehman et al., 2021*; gemcitabine against pancreatic cancer *Liu et al., 2022*; and carboplatin against high-grade serous ovarian carcinoma (*du Manoir et al., 2022*; *Figure 5H*)). Moreover, enrichment of *FYN* has also been observed in residual disease following chemotherapy, including neoadjuvant chemostherapy plus bevacizumab-treated HER2 negative breast cancer (*Kimbung et al., 2018*), neoadjuvant chemoradiotherapy combined plus atezolizumab-treated esophageal cancer (*van den Ende et al., 2021*), and chemoradiotherapy-treated colorectal cancer (*Snipstad et al., 2010*), indicating its potential role in mediating drug tolerance during chemotherapy (*Figure 5I*). While the causal relationship between FYN expression drug tolerance in response to various therapeutic interventions warrants further study, these evidence suggest that FYN expression is associated with drug tolerance. Notably, an analogous increase in *KDM4* was not consistently observed across all tumor models tested in *Figure 5H and I* (*Figure 5—figure supplement 1A and B*). This suggests that, as previously noted in *Figure 5A*, while *FYN* serves as a general mediator of drug tolerance, the specific mechanisms underlying its upregulation may vary depending on the cancer type and the drug being administered. Taken together, these lines of evidence further support our findings in TNBC cell lines and suggest that *FYN* acts as a common mediator of drug tolerance.

## Discussion

In this study, we employed combinatorial CRISPR screening to identify combinations of TKIs that exhibit synergistic effects in eliminating TNBC. We discovered and validated that concurrent targeting of FYN, along with other tyrosine kinases such as IGF1R, EGFR, or ABL2 can synergistically eradicate TNBC and impede cancer growth. Our findings also provide evidence that the transcriptional upregulation of *FYN*, facilitated by the activation of KDM4 histone demethylases, confers resistance and persistence to TKIs. Upregulation of *FYN* is a general feature of drug-tolerant cancer cells, suggesting the association of *FYN* expression with drug resistance and tumor recurrence after treatment.

This research provides basis for breakthrough combinatorial therapy achieving effective targeted therapy with minimal risk of developing resistance. Our combinatorial CRISPR screening demonstrates that treatment with TKIs or histone demethylase inhibitors enhances the sensitivity of cells to other TKIs. Consequently, drug combinations exhibit a more potent inhibition of cancer growth than the simple sum of the therapeutic effects of individual drugs. Furthermore, synergistic drug combinations enable a reduction in the dosage of each drug, with minimal compromise in therapeutic efficacy. Such combinations yield a therapeutic response comparable to that achieved with significantly higher doses of each individual agent. We anticipate that combinatorial therapy has the potential to mitigate side effects by minimizing the dosage of each drug, thus widening the therapeutic window. Further studies should elucidate the downstream mechanisms by which FYN upregulation contributes to drug tolerance. SRC family kinases are known to upregulate multiple signaling pathways including ERK, AKT and p38 pathway (*Zhang and Yu, 2012*). Although our study showed that, at least in MDA-MB-231 cell line, FYN depends on p38 activity for TKI resistance (*Figure 2J–L*), it should further be shown while this downstream mechanism is generalizable in other cancers. The downstream mechanism of p38 that contributes to drug resistance would also be of great interest to identify novel therapeutic approaches minimizing drug tolerance. Also, our combinatorial CRISPR screen results warrant further studies with other synthetic lethal gene combinations other than those involving FYN that are not deeply investigated in this study. Particularly, FGFR2, TEK, FRK identified as key nodes in *Figure 2C*

may be of particular interest as they are also associated with cancer cell survival (*Formisano et al., 2019*; *Zhang et al., 2020*).

It is intriguing to observe that *FYN* is specifically upregulated in various models of drug resistance and tolerance. SRC family kinases, which includes FYN, have been linked to drug resistance (*Zhang and Yu, 2012*; *Girotti et al., 2013*). In line with this, phosphoproteomic analysis of neoadjuvant chemotherapy resistant TNBC patient derived xenografts showed upregulated SRC family kinase networks including kinases and their substrates and adaptors (*Kohale et al., 2022*). Our findings reveal that *FYN* is specifically upregulated at the mRNA level possibly through epigenetic regulations, providing further depth to our understanding of drug resistance in cancer therapy. The epigenetic reprogramming of the drug tolerant cells may be distinct depending on the tumor type or the therapeutic interventions, as KDM4A, which we show is increased upon TKI treatment in TNBC cell line in *Figure 3*, is not significantly regulated in other cancers analyzed in *Figure 5*. Therefore, the context-dependent mechanisms underlying FYN upregulation, and its essentiality in constituting drug tolerance remains as subjects for further study.

Furthermore, our work highlights the significance of histone demethylases in TKI resistance. Numerous histone demethylases have been implicated in drug resistance and tolerance across different cancer drug types. For instance, the KDM5 family of H3K4 demethylases has been associated with the drug-tolerant persister phenotype against multiple TKIs (*Sharma et al., 2010*). In our study, we identify *KDM4* as a critical factor in the generation of drug-tolerant persisters in breast cancer. *KDM4* is known to be upregulated in various cancers, including breast cancer, and promotes key malignant traits. Previous studies have demonstrated the essential role of KDM4 in induced pluripotency through its interaction with pluripotency factors (*Loh et al., 2007*). These findings suggest that an KDM4 inhibitor could be a promising therapeutic target with specific activity against cancer stem cells. Consistently, specific inhibitors targeting KDM4 have recently been developed and shown to inhibit the generation of breast cancer stem cells (*Metzger et al., 2017*). The mechanisms underlying KDM4 upregulation upon drug treatment is currently unclear and remains as subjects for further study. Nevertheless, given our discoveries regarding the involvement of *KDM4* in drug resistance in breast and lung cancer, the development of novel drugs targeting KDM4 holds significant therapeutic potential.

## Materials and methods

### Cell culture

HEK293T, MDA-MB-231, Hs578T cells were obtained from American Type Cell Culture (ATCC) HCC1143, HCC1395, HCC1937 were obtained from Korean cell line bank. They are validated with STR profiling. HEK293T and MDA-MB-231 were grown in DMEM supplemented with 10% FBS and penicillin/streptomycin. Hs578T and HCC1143, HCC1395, HCC1937 were grown in RPMI1640 supplemented with 10% FBS. Parental and gefitinib-, and Osimertinib- resistant PC9 and HCC827 cells were kind gifts from Jae Cheol Lee (Asan Medical Center, Seoul, Korea) and were grown in RPMI1640 supplemented with 10% FBS. Gefitinib and osimertinib resistant cells are maintained in the presence of 1 μM gefitnib and 0.5 μM osimertinib, respectively. Gefitinib-resistant derivatives of PC9 and HCC827 cells were generated as described previously by treating cells with escalating dose of gefitinib (*Zhang et al., 2012*). All cell culture medium and supplements are purchased from Welgene Inc.

### Combinatorial library construction

Combinatorial library was constructed as previously described (*Wong et al., 2015*). The sgRNAs used in the screens were cloned in pAWp28 storage vector in two versions: one version containing human U6 driven sgRNA with wild type scaffold, and another containing mouse U6 driven sgRNA with cr2 variant scaffold. The sgRNA expression cassette consisting of U6 promoter and sgRNA were subject to one-pot, iterative cloning into lentiviral pTK799 vector using BglII-MfeI restriction sites flanking the sgRNA expression cassette and BamHI-EcoRI sites in pTK799. pTK799 vector is derived from pAWp12 (*Wong et al., 2015*) by replacing CMV-GFP selectable marker to EFS-Puro.

## Combinatorial CRISPR screening procedure

Lentivirus was generated in HEK293T cells by transfecting lentiviral transfer vector, and helper vectors (psPAX2, and pVSV-G) using Fugene HD (Promega). Lentiviral supernatant was collected 48 hr after transfection, and was frozen and stored in –80 C. The appropriate titer for lentiviral transduction was determined by transducing MDA-MB231 cells with two-fold serial dilution of lentiviral supernatant, selecting with puromycin 2 days after transduction for 2 days, and determining cell viability with AQuaeous one cell viability, MTS assay (Promega). After determining the titer of lentiviral supernatant, 100 million MDA-MB231 cells carrying constitutively expressed Cas9 were transduced with CombiGEM library at MOI of 0.3. The expected initial coverage is 100 million x 0.3/ (54,289 different sgRNA combinations)=553. Three days after transduction, the cells were either harvested as day 0 sample or selected with 2 µg/mL puromycin (Invitrogen). Cells were treated with benzonase before harvesting to minimize carryover of plasmid DNA in lentiviral supernatant. Cells were grown in the presence of 2 µg/mL puromycin for 20 days before harvesting.

The genomic DNA of harvested cells were isolated using Blood & Cell Culture Maxi kit (QIAGEN). The PCR amplicon spanning the two sgRNAs were generated with PCR using Q5 High Fidelity DNA polymerase (New England Biolabs) and the following primers:

> F: CAAGCAGAAGACGGCATACGAGAT CCTAGTAACTATAGAGGCTTAATGTGCG
> R: AATGATACGGCGACCACCGAGATCTACAC NNNNNN ACACGAATTCTGCCGTGGATCCA A

The six nucleotides described as 'NNNNNN' in reverse primer above represents unique index to identify biological replicates in multiplexed NGS run.

The PCR protocol involves 60 s of initial DNA denaturation at 98 °C, and 20 cycles of 10 s denaturation at 98 °C, 10 s annealing at 67 °C, and 120 s elongation at 72 °C. All genomic DNA isolated were used in PCR reaction at concentration of 40 µg/mL. All PCR products were combined and precipitated with isopropanol at room temperature. The precipitated DNA was resuspended in 400 µL EB buffer (QIAGEN) and gel purified. The purified PCR products were sent for NGS by NextSeq500 paired end sequencing with the following sequencing primers:

> Forward read: GGACTAGCCTTATTTGAACTTGCTATGCAGCTTTCTGCTTAGCTCTCAAAC
> Forward index read: CGGTGCCACTTTTTCAAGTTGATAACGGACTAGCCTTATTTTAACTTG CTATTTCTAGCTCTAAAAC
> Reverse read: GCA CCG AGT CGG TGC TTT TTT GGA TCC ACG GCA GAA TTC GTGT

The raw GI scores calculated as deviation from the quadratic fit of the expected-observed Z score plot. The GI scores were normalized by dividing the raw GI scores with the standard deviation of the GI scores obtained from the 200 nearest neighbors in terms of expected Z scores (*Figure 1—figure supplement 1I*; *Han et al., 2017*).

## Data analysis

The sgRNA sequences were identified and their occurrences were counted with C++ script deposited in GitHub (copy archived at *Kim, 2025*).

## Validation of screens using sgRNAs

Individual sgRNA was cloned to either pTK1329, and pTK1336 that are both derived from pAWp12 with EFS-GFP and EFS-mCherry, respectively, as selectable markers. Validation of synthetic lethality between gene A and B were analyzed by transducing MDA-MB-231 Cas9 cells with four combinations of lentiviral supernatant pairs (MOI ~0.5 each) containing (*Wu et al., 2016*) GFP-sgA and mCherry-sgB; (*Hanahan and Weinberg, 2011*) GFP-sgA and mCherry-sgCon; (*Li et al., 2021*) GFP-sgCon and mCherry-sgB; (*Litzenburger et al., 2011*) GFP-sgCon and mCherry-sgCon. The fraction of GFP/mCherry double positive cells were analyzed using BD Accuri C6 and its accompanying software. The expected fold change in sgA +sgB were calculated as $FC_{sgA+sgCon}$ x $FC_{sgCon+sgB}$, where FC is normalized fold change in fraction of GFP/mCherry double positive cells relative to those transduced with GFP-sgCon and mCherry-sgCon. The sgRNAs used for this study are listed in *Supplementary file 3*.

## MTT cell viability assay

Cells were seeded at 1000–2000 cells/well in 96-well plate. Tyrosine kinase inhibitors at indicated combination of dose were treated 12 hr after seeding, and cells were grown for 3 days. The relative viability was measured by EzCytox cell viability assay (Dojindo). The absorbance at 450 nm wavelength was measured using EnVision multimode plate reader (PerkinElmer).

## Cell death and cell proliferation assay

Cells were incubated with tyrosine kinase inhibitors for 48 hr. Cell proliferation was quantified with BrdU assay using FITC conjugated BrdU antibody (Biolegend, 364103) and propidum iodide/RNase A solution (Cell Signaling), analyzed with BD Accuri C6 and accompanied software. Cell death was quantified with Live-Dead cell staining kit (Molecular Probes) by flow cytometry analysis using BD Accuri C6 and accompanied software.

## Western blot analysis

Cells were treated with drugs for 48 hr unless otherwise indicated. Cells were lysed in RIPA buffer (50 mM Tris-HCl pH 7.5, 150 mM sodium chloride, 0.1% sodium dodecyl sulfate, 0.5% sodium deoxycholate, 1% NP-40, 1 mM EDTA) supplemented with protease inhibitor (aprotinin, leupeptin, pepstatin A, and phenylmethylsulfonyl fluoride [PMSF]) and phosphatase inhibitor (sodium fluoride and sodium orthovanadate) cocktail. Antibodies used for western blot analysis were: anti-SRC (Santa Cruz biotechnology, mouse sc-5266), anti-FYN (Cell Signaling Technology, rabbit #4023), anti-phospho-FYN Y530 (Invitrogen, PA5-36644), anti-GAPDH (Cell Signaling Technology, rabbit #5174), anti-KDM4A (Bethyl Laboratories, rabbit A300-861A), anti-phospho-ERK T202/Y204 (Cell Signaling Technology, rabbit #9101), anti-ERK (Cell Signaling Technology, rabbit #9102), anti-phospho-STAT3 Y705 (Cell Signaling Technology, rabbit #9145), anti-STAT3 (Cell Signaling Technology, rabbit #4904), anti-phospho-AKT (Cell Signaling Technology, mouse #4051), anti-AKT (Cell Signaling Technology, rabbit #9272), anti-phospho-p38 (Cell Signaling Technology, rabbit #9215), anti-p38 (Cell Signaling Technology, rabbit #8690).

## RT-qPCR analysis

RNA is extracted from cultured cells with Trizol (Invitrogen) according to the manufacturer's instructions. The precipitated RNA pellet was resuspended in RNase-free water (Enzynomics), and was subject to reverse transcription with M-MLV reverse transcriptase (Enzynomics) at 37 °C for 2 hr, followed by heat inactivation at 95 °C for 5 min. The resulting cDNA was used for quantitative real time PCR using TOPreal SYBR Green qPCR premix (low ROX, Enzynomics) reagent and CFX96 Real-Time PCR detection system (Bio-Rad).

## Xenograft assay

All animal experiments were approved by IACUC of Korea Institute of Science and Technology (KIST) and conforms to ARRIVE guideline. Six-week-old female nude mice were injected with $5 \times 10^6$ MDA-MB-231 cells suspended in 1:1 (w/w) mixture of PBS and growth factor reduced Matrigel (Corning) in fourth inguinal mammary fat pad. Starting 2 weeks after tumor cell injection, saracatinib (50 mg/kg mouse body weight, MedChemExpress), NVP-ADW742 (20 mg/kg, Sigma), gefitinib (20 mg/kg, MedChemExpress), QC6352 (10 mg/kg, MedChemExpress) in 45% saline +40% polyethyleneglycol 300 (Sigma) +5% Tween-80 (Sigma) +5% DMSO (Sigma) were injected intraperitoneally every 24 hr for 2 weeks. Tumor volume was measured by digital caliper and calculated as (width)$^2$ x length x 0.5.

## Public database analyses

Gene Expression Omnibus (GEO) data with breast cancer cohort (GSE25066 *Hatzis et al., 2011*) were analyzed using web-based platform Cancer Target Gene Screening (http://ctgs.biohackers.net/; *Kim et al., 2020*). Cancer Cell Line Encyclopedia (CCLE) data were analyzed using depmap R package version 1.14. The list of GEO data used for analysis are listed in *Supplementary file 4*.

## Primary TNBC organoid culture and drug treatment

Specimens was obtained from enrolled patients with TNBC breast cancer with National Cancer Center IRB approval. Tumor tissue was collected and transferred using cold RPMI1640 media at the

National Cancer Center (Goyang, Republic of Korea). Then the tissues were dissected with a blade on Petri dish and enzymatically digested with dissociated kit (Multi Tissue Dissociation Kit 1, 130-110-201) by gentle MACS Dissociators (Miltenyi Biotec, Germany). After cell counting, $1 \times 10^5$ cells were embedded in 40 µl of Matrigel (Corning) and seeded into each well of a 24-well cell culture plate. After the matrigel was solidified, 500 µL organoid medium supplemented with advanced DMEM/F12 medium (Invitrogen, USA), B27 (Invitrogen, USA, 17504), 1.25 mM N-acetylcysteine (Sigma, USA, A9165), 5 ng/mL EGF (PeproTech, USA, AF-100–15), 20 ng/mL FGF-10 (PeproTech, 100–26), 5 ng/mL FGF7 (PeproTech, AF-100–19), 50 ng/mL R-spondin 1 (Qkine, Qk006), 5 mM nicotinamide (Sigma, N0636), 500 nM A83-01 (Tocris, 2939), 1 X GlutaMAX (Gibco, 35050–061), 100 µg/mL primocin (Invivogen, ant-pm), 10 mM HEPES (Gibco, 15630–080), 1 X Noggin (U-Protein Express BV, N002), 1 X ITS (Gibco, 12585014), 100 nM β-estradiol (Sigma, E2758), 1 µg/mL Hydrocortisone (Sigma, H0888), 5 nM Heregulin (PeproTech, AF-100–03), 500nM SB202190 (R&D system, 1264), 10 µM Y27632 (TOCRIS, 1254) as described by Clevers and colleagues (*Sachs et al., 2018*), was added to each well and the cells grown under standard culture conditions (37 °C, 5% $CO_2$).

## Acknowledgements

We thank BioMicro Center of MIT for sequencing analysis. We thank Jae Cheol Lee (Asan Medical Center, Korea) for providing gefitinib resistant lung cancer cell lines. This work was supported by Korea Institute of Science and Technology (KIST) Institutional Programs (2E32331 to TK); and National Research Foundation of Korea, funded by the Korean government (MSIT) (2021R1A2C1093499 to TK, 2020 M3A9A5036362 to SYK).

## Additional information

### Funding

| Funder | Grant reference number | Author |
| --- | --- | --- |
| National Research Foundation of Korea | 2021R1A2C1093499 | Tackhoon Kim |
| Korea Institute of Science and Technology | 2E32331 | Tackhoon Kim |
| National Research Foundation of Korea | 2020M3A9A5036362 | Sun-Young Kong |

The funders had no role in study design, data collection and interpretation, or the decision to submit the work for publication.

### Author contributions

Tackhoon Kim, Conceptualization, Formal analysis, Supervision, Funding acquisition, Writing – original draft, Writing – review and editing; Byung-Sun Park, Formal analysis, Investigation; Soobeen Heo, Donghwa Kim, Sang Kook Lee, Methodology; Heeju Jeon, Formal analysis, Visualization; Jaeyeal Kim, Data curation; So-Youn Jung, Sun-Young Kong, Supervision, Methodology; Timothy Lu, Conceptualization, Supervision, Funding acquisition

### Author ORCIDs

Tackhoon Kim ⬤ https://orcid.org/0000-0001-7911-6273

### Ethics

All experiments with human tumor organoids were conducted in accordance with the requirements of the National Cancer Center Institutional Review Board (IRB). Informed consent and consent to publish were obtained from patients.
All animal experiments were approved by IACUC of Korea Institute of Science and Technology (KIST).

Reviewer #1 (Public review): https://doi.org/10.7554/eLife.93921.3.sa1
Author response https://doi.org/10.7554/eLife.93921.3.sa2

## Additional files

### Supplementary files
Supplementary file 1. Tyrosine kinases subject to CRISPR screens in this study.

Supplementary file 2. Growth phenotype scores Z and gene interaction scores GI calculated with combinatorial screens in *Figure 1*.

Supplementary file 3. List of sgRNA used in the study.

Supplementary file 4. List of GEO data used for analysis.

MDAR checklist

Source data 1. Source data for *Figure 2F*, *Figure 3J* and *Figure 2—figure supplement 3B*.

### Data availability
The NGS data for CRISPR screening results are available under NCBI SRA accession code PRJNA976939. The C++script for CRISPR screen analysis is deposited in GitHub (copy archived at *Kim, 2025*).

The following dataset was generated:

| Author(s) | Year | Dataset title | Dataset URL | Database and Identifier |
|-----------|------|---------------|-------------|--------------------------|
| Kim T | 2025 | Combinatorial CRISPR screens reveal synthetic lethal tyrosine kinase inhibitions | https://www.ncbi.nlm.nih.gov/bioproject/PRJNA976939 | NCBI BioProject, PRJNA976939 |

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
