## [Editor Report · eLife Assessment]

This study presents a **valuable** finding that synthetically lethal kinase genes FYN and KDM4 may play a role in drug resistance to kinase inhibitors in TNBC. The evidence supporting the claims of the authors is **solid**, although the exploration of the upstream mechanisms regulating KDM4A or the downstream pathways through which FYN upregulation confers drug resistance would have strengthened the study. The work will be of interest to medical biologists working in the field of breast cancer.

---

## [Referee Report · Reviewer #1 (Public review)]

Summary:

The authors employed a combinatorial CRISPR-Cas9 knockout screen to uncover synthetically lethal kinase genes that could play a role in drug resistance to kinase inhibitors in triple-negative breast cancer. The study successfully reveals FYN as a mediator of resistance to depletion and inhibition of various tyrosine kinases, notably EGFR, IGF-1R, and ABL, in triple-negative breast cancer cells and xenografts. Mechanistically, they demonstrate that KDM4 contributes to the upregulation of FYN and thereby is an important mediator of the drug resistance. All together, these findings suggest FYN and KDM4A as potential targets for combination therapy with kinase inhibitors in triple-negative breast cancer. Moreover, the study may also have important implications for other cancer types and other inhibitors, as the authors suggest that FYN could be a general feature of drug-tolerant persister cells.

Strengths:

(1) The authors used a large combination matrix of druggable tyrosine kinase gene knockouts, enabling studying of co-dependence of kinase genes. This approach mitigates off-target effects typically associated with kinase inhibitors, enhancing the precision of the findings.

(2) The authors demonstrate the importance of FYN in drug resistance in multiple ways. They demonstrate synergistic interactions using both knockouts and inhibitors, while also revealing its transcriptional upregulation upon treatment, strengthening the conclusion that FYN plays a role in the resistance.

(3) The study extends its impact by demonstrating the potent in vivo efficacy of certain combination treatments, underscoring the clinical relevance of the identified strategies.

Weaknesses:

(1) The combination of FYN knockout with other gene knockouts exhibits only very modest synergy. The high standard deviation observed for FYN knockout in Figure S2A weakens the robustness of these findings. As combination treatments involving inhibitors did demonstrate stronger synergistic effects, the data still support the role of FYN in regulating sensitivity to the described drugs.

(2) While the study identifies KDM4A as a key contributor to FYN upregulation, it does not fully explore the upstream mechanisms regulating KDM4A or the downstream pathways through which FYN upregulation confers drug resistance. These unaddressed questions limit the mechanistic understanding that can be obtained from this study.

(3) FYN has been implicated in drug resistance in previous studies, and other mechanisms for its upregulation and downstream effects have already been described. While this study adds value to the existing literature in the context of breast cancer, it does not present entirely novel findings regarding FYN's role in drug resistance.

---

## [Author Response]

The following is the authors’ response to the original reviews.

**Public Reviews:**

**Reviewer #1 (Public Review):**
Summary:The authors employed a combinatorial CRISPR-Cas9 knockout screen to uncover synthetically lethal kinase genes that could play a role in drug resistance to kinase inhibitors in triple-negative breast cancer. The study successfully reveals FYN as a mediator of resistance to depletion and inhibition of various tyrosine kinases, notably EGFR, IGF-1R, and ABL, in triple-negative breast cancer cells and xenografts. Mechanistically, they demonstrate that KDM4 contributes to the upregulation of FYN and thereby is an important mediator of drug resistance. All together, these findings suggest FYN and KDM4A as potential targets for combination therapy with kinase inhibitors in triple-negative breast cancer. Moreover, the study may also have important implications for other cancer types and other inhibitors, as the authors suggest that FYN could be a general feature of drug-tolerant persister cells.Strengths:(1) The authors used a large combination matrix of druggable tyrosine kinase gene knockouts, enabling studying of co-dependence of kinase genes. This approach mitigates off-target effects typically associated with kinase inhibitors, enhancing the precision of the findings.(2) The authors demonstrate the importance of FYN in drug resistance in multiple ways. They demonstrate synergistic interactions using both knockouts and inhibitors, while also revealing its transcriptional upregulation upon treatment, strengthening the conclusion that FYN plays a role in the resistance.(3) The study extends its impact by demonstrating the potent in vivo efficacy of certain combination treatments, underscoring the clinical relevance of the identified strategies.Weaknesses:(1) The methods and figure legends are incomplete, posing a barrier to the reproducibility of the study and hindering a comprehensive understanding and accurate interpretation of the results.

We thank the reviewer for pointing this out. We tried adding as much detail in methods and figures legends as possible to maximize reproducibility and accuracy in interpreting our results as will be described for our responses for the recommendations for authors.

(2) The authors make use of a large quantity of public data (Fig. 2D/E, Fig. 3F/L/M, Fig 4C, Fig 5B/H/I), whereas it would have strengthened the paper to perform these experiments themselves. While some of this data would be hard to generate (e.g. patient data) other data could have been generated by the authors. The disadvantage of the use of public data is that it merely comprises associations, but does not have causal/functional results (e.g. FYN inhibition in the different cancer models with various drugs). Moreover, by cherry-picking the data from public sources, the context of these sources is not clear to the reader, and thus harder to interpret correctly. For example, it is not directly clear whether the upregulation of FYN in these models is a very selective event or whether it is part of a very large epigenetic re-programming, where other genes may be more critical. While some of the used data are from well-known curated databases, others are from individual papers that the reader should assess critically in order to interpret the data. Sometimes the public data was redundant, as the authors did do the experiments themselves (e.g. lung cancer drug-tolerant persisters), in this case, the public data could also be left out.More importantly, the original sources are not properly cited. While the GEO accession numbers are shown in a supplementary table, the articles corresponding to this data should be cited in the main text, and preferably also in the figure legend, to clarify that this data is from public sources, which is now not always the case (e.g. line 224-226). If these original papers do already mention the upregulation of FYN, and the findings from the authors are thus not original, these findings should be discussed in the Discussion section instead of shown in the Results.

We welcome the reviewer’s concern. As reviewer pointed out, our analysis with FYN expression levels in multiple studies with drug tolerant cells may merely reflect association and not causal relationships. We had at least shown that FYN inhibition may reduce drug tolerance in TNBC and EGFR inhibitor treated lung cancer cells (figures 2H, 5E). The causal role of FYN in emergence of drug tolerance in other cancers treated with different drugs (such as irinotecan treated colon adenocarcinoma and gemcitabine treated pancreatic adenocarcinoma) may be beyond scope of this study. We made a brief discussion addressing this concern in lines 273-275.

We also added proper citations of the public data used in this study in main text and figure legends in lines 267-269. The GEO accession numbers are listed in supplementary table S2. Importantly, none of the referenced studies identified FYN as key factor in generating drug tolerant cells.

(3) The claim in the abstract (and discussion) that the study "highlights FYN as broadly applicable mediator of therapy resistance and persistence", is not sufficiently supported by the results. The current study only shows functional evidence for this for an EGFR, IGF1R, and Abl inhibitor in TNBC cells. Further, it demonstrates (to a limited extent) the role of FYN in gefitinib and osimertinib resistance (also EGFR inhibitors) in lung cancer cells. Thus, the causal evidence provided is only limited to a select subset of tyrosine kinase inhibitors in two cancer types. While the authors show associations between FYN and drug resistance in other cancer types and after other treatments, these associations are not solid evidence for a causal connection as mentioned in this statement. Epigenetic reprogramming causing drug resistance can be accompanied by altered gene expression of many genes, and the upregulation of FYN may be a consequence, but not a cause of the drug resistance. Therefore, the authors should be more cautious in making such statements about the broad applicability of FYN as a mediator of therapy resistance.

We fully agree with the reviewer’s concern that FYN upregulation is simply an association, and may not be the cause of drug tolerance and resistance. Therefore, to accurately convey our findings, we edited our manuscript in lines 34-36 in abstract to “*FYN* expression is associated with therapy resistance and persistence by demonstrating its upregulation in various experimental models of drug-tolerant persisters and residual disease following targeted therapy, chemotherapy, and radiotherapy” and lines 288-290 in discussion to “ Upregulation of *FYN* is a general feature of drug tolerant cancer cells, suggesting the association of *FYN* expression with drug resistance and tumor recurrence after treatment.” We hope this satisfies the reviewer.

(4) The rationale for picking and validating FYN as the main candidate gene over other genes such as FGFR2, FRK2, and TEK is not clear.a. While gene pairs containing FGFR2 knockouts seemed to be equally effective as FYN gene pairs in the primary screening, these could not be validated in the validation experiment. It is unclear whether multiple individual or a pool of gRNAs were used for this validation, or whether only 1 gRNA sequence was picked per gene for this validation. If only 1 gRNA per gene was used, this likely would have resulted in variable knockout efficiencies. Moreover, the T7 endonuclease assay may not have been the best method to check knockout efficiency, as it only implies endonuclease activity around a gene (but not to the extent of indels that can cause frameshifts, such as by TIDE analysis, or extent of reduction in protein levels by western blot).b. Moreover, FRK2 and TEK, also demonstrated many synergistic gene pairs in the primary screen. However, many of these gene pairs were not included in the validation screening. The selection criteria of candidate gene pairs for validation screening is not clear. Still, TEK-ABL2 was also validated as a strong hit in the validation screen. The authors should better explain the choice of FYN over other hits, and/or mention that TEK and FRK2 may also be important targets for combination treatment that can be further elucidated.

We thank the reviewer for improving our manuscript. We had concerns with the generalizability of FGFR2, FRK and TEK in TNBC as their expressions are very low in MDA-MB-231, nor were they enriched in TNBC compared to cancer cell lines of other subtypes. We added a brief comment on this concern in results section and discussion section (lines 150-154, figure S3). Although we acknowledge that the validations done in figure 2B is a result of only one guide RNA, with validations with pharmacological inhibition of FYN (figure 2F-I), we hope the reader and reviewer can be convinced with our key findings in synthetic lethality between FYN and other tyrosine kinases.

(5) On several occasions, the right controls (individual treatments, performed in parallel) are not included in the figures. The authors should include the responses to each of the single treatments, and/or better explain the normalization that might explain why the controls are not shown.a. Figure 2G: The effect of PP2 treatment, without combined treatment, is not shown.b. Figure 2H/3G: The effect of the knockouts on growth alone, compared to sgGFP, is not demonstrated. It is unclear whether the viability of knockouts is normalized to sgGFP, or to each untreated knockout.c. Figure 2L: The effect of SB203580 as a single treatment is not shown.

We thank the reviewer for pointing this out. The data shown for all figures listed in these concerns were normalized by the changes in viability by pharmacological or genetic perturbations that synergized with TKIs (NVP-ADW742, gefitinib…etc.) used in the figures in the original manuscript. As reviewer had suggested, we newly added the effect of SB203580 and PP2 treatment on cell viability in supplementary figures S4A, S4K. SB203580 had no significant effect on cell viability, while PP2 treatment caused significant decrease in cell viability, which is expected as PP2 can inhibit activity of multiple Src family kinases. Regardless of the effect of SB203580 and PP2 on cell viability as single agent, it is evident that treatment of TKIs synergistically decreased cell viability in cancer cell lines. The change in viability by FYN or histone lysine demethylase knockout was also provided in newly added figure S4D and S6C. Notably, genetic ablation of FYN or histone lysine demethylases had modest, if any, influences on cell viability.

(6) The study examines the effects at a single, relatively late time point after treatment with inhibitors, without confirming the sequential impact on KDM4A and FYN. The proposed sequence of transcriptional upregulation of KDM4A followed by epigenetic modifications leading to FYN upregulation would be more compellingly supported by demonstrating a consecutive, rather than simultaneous, occurrence of these events. Furthermore, the protein level assessment at 48 hours (for RNA levels not clearly described), raises concerns about potential confounding factors. At this late time point, reduced cell viability due to the combination treatment could contribute to observed effects such as altered FYN expression and P38 MAPK phosphorylation, making it challenging to attribute these changes solely to the specific and selective reduction of FYN expression by KDM4A.

We thank the reviewer for pointing this out. We performed time course experiment for NVP-ADW742 treatment on MDA-MB-231 cells in our newly added figure 3E. Surprisingly, treatment of NVP-ADW742 increased KDM4A protein level within two hours. FYN protein accumulation followed KDM4A accumulation after 24 hours. This observation, with our chromatin immunoprecipitation data in figure 3O, provide evidence that FYN accumulation is a consequence of KDM4A accumulation and H3K9me3 demethylation upon TKI treatment. We newly discussed this data in results and discussion section in lines 214-216.

(7) The cut-off for considering interactions "synergistic" is quite low. The manual of the used "SynergyFinder" tool itself recommends values above >10 as synergistic and between -10 and 10 as additive (https://synergyfinder.fimm.fi/synergy/synfin_docs/). Here, values between 5-10 are also considered synergistic. Caution should be taken when discussing those results. Showing the actual dose response (including responses to each single treatment) may be required to enable the reader to critically assess the synergy, along with its standard deviation.

We thank the reviewer for careful comments. We reanalyzed our data with SynergyFinder plus tool (Zheng, Genomics, Proteomics, and Bioinformatics 2022), which implements mathematical models distinct from SynergyFinder 3, for more faithful implementation of Bliss, Loewe independence models, and more critically, calculates statistical significance of the synergy. We provide updates synergy plots with statistics in figures 2F, 3J, and S4B. All drug combinations show statistically significant synergy (p<0.01). We also add raw data used to calculate synergy in figures 2F, 3J and S4B in supplementary dataset S2.

(8) As the effect size on Western blots is quite limited and sometimes accompanied by differences in loading control, these data should be further supported by quantifications of signal intensities of at least 3 biological replicates (e.g. especially Figure 3A/5A). The figure legends should also state how many independent experiments the blots are representative of.

We added quantifications for figure 3A and 5A for better depiction of our results. Figure legends were edited to indicate this is a representative of three independent experiments.

(9) While the article provides mechanistic insights into the likely upregulation of FYN by KDM4A, this constitutes only a fragment of the broader mechanism underlying drug resistance associated with FYN. The study falls short in investigating the causes of KDM4A upregulation and fails to explore the downstream effects (except for p38 MAPK phosphorylation, which may not be complete) of FYN upregulation that could potentially drive sustained cell proliferation and survival. These omissions limit the comprehensive understanding of the complete molecular pathway, and the discussion section does not address potential implications or pathways beyond the identified KDM4A-FYN axis. A more thorough exploration of these aspects would enhance the study's contribution to the field.

We welcome the reviewer’s careful concern. We agree our delineation of mechanisms underlying TKI resistance in TNBC involving KDM4 and FYN is far from complete. The increases in expression of histone demethylases were observed in cancers treated with different drugs. The mechanisms governing the increase in histone demethylase expression is not known and is beyond the scope of this paper. We newly added this in discussion section in lines 299-304.

(10) FYN has been implied in drug resistance previously, and other mechanisms of its upregulation, as well as downstream consequences, have been described previously. These were not evaluated in this paper, and are also not discussed in the discussion section. Moreover, the authors did not investigate whether any of the many other mechanisms of drug resistance to EGFR, IGF1R, and Abl inhibitors that have been described, could be related to FYN as well. A more comprehensive examination of existing literature and consideration of alternative or parallel mechanisms in the discussion would enhance the paper's contribution to understanding FYN's involvement in drug resistance.

FYN has been implicated in TKI resistance in CML cell lines (Irwin, Oncotarget, 2015). In this study, FYN is similarly transcriptionally upregulated in imatinib resistant CML, and this upregulation is dependent on EGR1 transcription factor. To address this concern, we generated EGR1 KO MDA-MB-231 cells and tested whether these cells retain the ability to accumulate FYN. Consistent with the previous study, imatinib treatment increased EGR1 protein level. However, EGR1 knockout did not influence FYN accumulation in MDA-MB-231 cells. EGR1 mediated accumulation of FYN may be context specific phenomenon to CML (Figure S5B). We newly discussed this result in result sections in lines 187-190. We also acknowledge that SRC family kinases are generally involved in drug resistance in many cancers. We discuss the recent findings regarding SRC family kinases in drug resistance in result section in lines 145-147 and discussion sections in lines 315-317.

**Reviewer #2 (Public Review):**
Summary:Kim et al. conducted a study in which they selected 76 tyrosine kinases and performed CRISPR/Cas9 combinatorial screening to target 3003 genes in Triple-negative breast cancer (TNBC) cells. Their investigation revealed a significant correlation between the FYN gene and the proliferation and death of breast cancer cells. The authors demonstrated that depleting FYN and using FYN inhibitors, in combination with TKIs, synergistically suppressed the growth of breast cancer tumor cells. They observed that TKIs upregulate the levels of FYN and the histone demethylase family, particularly KDM4, promoting FYN expression. The authors further showed that KDM4 weakens the H3K9me3 mark in the FYN enhancer region, and the inhibitor QC6352 effectively inhibits this process, leading to a synergistic induction of apoptosis in breast cancer cells along with TKIs. Additionally, the authors discovered that FYN is upregulated in various drug-resistant cancer cells, and inhibitors targeting FYN, such as PP2, sensitize drug-resistant cells to EGFR inhibitors.Strengths:This study provides new insights into the roles and mechanisms of FYN and KDM4 in tumor cell resistance.Weaknesses:It is important to note that previous studies have also implicated FYN as a potential key factor in drug resistance of tumor cells, including breast cancer cells. While the current study is comprehensive and provides a rich dataset, certain experiments could be refined, and the logical structure could be more rigorous. For instance, the rationale behind selecting FYN, KDM4, and KDM4A as the focus of the study could be more thoroughly justified.
**Recommendations for the authors:**

**Reviewer #1 (Recommendations For The Authors):**
(1) The methods and figure legends are incomplete, posing a barrier to the reproducibility of the study and hindering a comprehensive understanding and accurate interpretation of the results. A critical revision of these aspects is needed, for example:a. Catalogue numbers of certain products critical to reproduce the study (e.g. antibodies) and/or at what company they have been purchased (e.g. used compounds)b. On several occasions the used concentrations of drugs or exposure time are not mentioned (e.g. Figure 2H, G (PP2), I, J, K, L, etc.)c. Figure legend of figure panels E-I in Figure 5 seems to be completely incorrect and not consistent with the figure axis etc.d. RT-qPCR methodology is not described in Methods.e. Western blot methods are very limited: these should be described in more detail or cite an article that does.f. Organoid culture: Information about the source of tumour cells (e.g. pre-treatment biopsy, material after surgery), isolation of tumor cells (e.g. methodology, characterization of material) and culture conditions (e.g. culture time before the experiment) is lacking.g. Information about how gefitinib/osimertinib-resistant PC9 and HCC827 cells are generated (as well as culture conditions and where they are from) is missing.

We thank the reviewer for pointing these out. We have done our best to add experimental details for reproducibility in methods section and figure legends in lines 343-348, 408-426, 431-432, 439-453, 648-650, 671-672 and 691-693.

(2) Figure 1B/C/D: it would be more meaningful if the most important hits (at least in one of these panels) were highlighted (e.g. line with gene-pair named), or visualized separately, so that the reader does not have to read the supplementary table to know what the most important hits were.

We thank the reviewer for careful concern. We newly added labels for key synergistic gene pairs in figures 1D as reviewer suggested.

(3) qPCR data shown in Figure S4 is from 1 independent experiment. As these experiments (especially qPCR) can be rather variable and the effect size is not very large, I would highly recommend repeating these experiments, or excluding them, as conclusions from them are not solid.

We found performing qPCR with many drugs that did not cause substantial synergistic cell death with NVP-ADW742 in figure S5C (figure S4A in previous version of manuscript) will not provide much additional insights. Also, as we were more interested in finding direct regulators of FYN expression, we focused on drugs that inhibit epigenetic regulator that activate transcription. Therefore, we focused on performing FYN qPCR with drug combinations involving GSK-J4 (KDM6 inhibitor) and pinometostat(DOT1L inhibitor). As shown in our newly added figure in S5D, while GSK-J4 inhibited FYN expression, pinometostat failed to do so. Also, we also confirm that knockout of KDM5 or KDM6 reproducibly failed to decrease FYN expression upon TKI treatment (figure S5E and S5G). The new results are discussed in lines 193-198. We hope these additions satisfy the reviewer.

(4) For validation of synergistic knockouts, it would be helpful for the interpretation to also show the viability/growth of each knockout (or treatment), instead of mostly normalized scores. For example, the reader now has no insight into whether FYN knockout itself already affects cell viability, or not. If it (or EGFR/IGF1R/ABL knockout) would already substantially affect cell viability, a further reduction in cell viability may not be as relevant as when it would not affect cell viability at all.

We thank the reviewer for pointing this out. We replaced our figure in figure 2A to indicate raw changes in cell viability in each single and double knockout cells in figure S2A. We hope this satisfies the reviewer.

(5) The curve fitting as in Figure 2G is somewhat misleading. While the curve seems to be forced to go from 1-0, the +PP2 dose-response curve does actually not seem to start at 1, but rather at 0.8, likely resulting from the effect of PP2 as a single treatment, thus, effects may be interpreted as more synergistic than that they truly are.

The results shown in figure 2G is actually normalized to cells treated or not with PP2 to better reflect the effect of NVP-ADW742, gefitinib and imatinib in the presence of PP2. So viability value starting at 0.8 is not because of the effect of PP2 treatment as single agent (because it is normalized to PP2 treated cells), but is actually because very small dose of particularly NVP-ADW742 resulted in modest decrease in viability. To more accurately depict our findings, we added the data point in figure 2G with TKI dose of 0uM at viability 1. We also added details for normalization of viability in figure legends.

(6) The readability of the paper could be enhanced by higher-quality images (now the text is quite pixelated).

We had technical difficulties in converting file types. We have replaced figures for better resolution for all main and supplementary figures.

(7) The discussion now contains one paragraph about the selectivity of kinase inhibitors, and that repurposing of inhibitors with more relaxed specificity or multi-kinase inhibitors can be beneficial. This does not seem to fall within the scope of the study, as there was no comparison between selective and non-selective inhibitors. It was also not clearly mentioned that the non-selective inhibitors worked better than the gene knockouts, or that for example, KDM3 and KDM4 knockout together worked better than only KDM4 knockout. It is recommended to either remove this paragraph, or rephrase it so that it better fits the actual results

We agree with the reviewer. We chose to remove this paragraph in lines 308-313.

(8) The entire paper does not discuss any known functions of FYN. Its function could be very briefly introduced in the results section when highlighting it as an important hit. More importantly, its known role in cancer and especially drug resistance should be discussed in the discussion (see also Public review).

We thank the reviewer for pointing this out. We added brief description of the role of FYN in cancer malignancy and drug resistance in lines 145-147. Particularly, FYN accumulation by EGR1 transcription factor had been described in the context of imatinib resistant chronic myeloid leukemia (Irwin, Oncotarget, 2015). To address this, we tested whether EGR1 knockout decreases FYN level in MDA-MB-231 (Figure S5A). Notably EGR1 knockout failed to decrease FYN protein level. This result was discussed in lines 187-190.

(9) Textual changes including:a. Line 29 (and others) "Massively parallel combinatorial CRISPR screens": I would rather choose a more descriptive term, such as "combinatorial tyrosine kinase knockout CRISPR screen", which already clarifies the screen used knockouts of (druggable) tyrosine kinases only. Using both "Parallel" and "combinatorial" is somewhat redundant, and "massively" is subjective, in my opinion.

Manuscript edited as suggested (lines 29, 63, 86, 283). The term “massively parallel” have been removed as they don’t significantly change our scientific findings.

b. Line 67 (and others): "to identify ... for elimination of TNBC": while this may be its potential implication, this study has identified genes in (mostly) TNBC cell lines and cell line xenografts. Please rephrase to something more within the scope of this research.

Manuscript edited as suggested (lines 68-69) as “we utilize CombiGEM-CRISPR technology to identify tyrosine kinase inhibitor combinations with synergistic effect in TNBC cell line and xenograft models for potential combinatorial therapy against TNBC.” We hope it satisfies the reviewer.

c. Line 31 (and others): Please check the capitals of words describing inhibitors, and make them consistent (e.g. Imatinib written with capital I, other inhibitors without capitals).

We thank the reviewer for catching this error. We changed all “imatinib” and “osimertinib” to lowercase.

d. Line 71: "... combining PP2, saracatinib (FYN inhibitor), .." ..." Here it is not clear PP2 is a FYN inhibitor, and, as saracatinib is a well-known Src-inhibitor, it is not correct to just say "FYN inhibitor". Better to rephrase to something such as: →"combining PP2 (Lck/Fyn inhibitor), saracatinib (Src/FYN inhibitor).

As reviewer noted, most Src family kinase inhibitors are not selective against specific member among other Src family members. Therefore, we changed line 73 to “PP2, saracatinib (Src family kinase / FYN inhibitor).”

e. Line 81: "The resulting library enabled massively parallel screens of pairwise knockouts, .." To clarify this is for the selected kinases only: "The resulting library enabled screens of pairwise knockouts of the 76 tyrosine kinase genes, .."

Manuscript edited as suggested by the reviewer in line 86.

f. Line 88 (and others): "after infection" consider rephrasing to "after transduction" as this is more commonly used when using lentiviral vectors only.

We thank the reviewer for this. Every “infection” that designates lentiviral transduction were changed to “transduction”.

g. Line 97-99: While being described as "good" correlation, a correlation of the same sgRNA pair, yet in a different order, of r=0.5 does not seem to be very good, neither does a correlation of r=0.74 for biological replicates. Please consider describing in a less subjective way.

We removed the subjective terms and changed the manuscript as follows: “sgRNA pair (e.g., sgRNA-A + sgRNA-B and sgRNA-B + sgRNA-A) were positively correlated (r = 0.50) and were combined when calculating Z (Fig. S1D). The Z scores for three biological replicates were also correlated with r = 0.74 between replicates #2 and #3 (Fig. S1E).” in lines 97-101.

h. Lines 92-96 and lines 102-115: The results section here contains quite a lot of technical information. While some information may be directly needed to understand the described results (such as a very short and simple explanation of how to interpret gene interaction score), other information may be more appropriate for the Methods section, to enhance the readability of the paper. Consider simplifying here and giving a more detailed overview in the Methods section. Also, the text is not entirely clear. You seem to give two separate explanations of how the GI scores were calculated (Starting in lines 106 and 111): please rephrase and clearly indicate the connections between those two explanations (in the Methods section).

We thank the reviewer for valuable suggestion. We moved significant portions of the technical descriptions in methods section. We also clarified the text regarding the procedures for calculating GI scores in lines 385-387.

i. Line 142: "These findings suggest that gene A could represent an attractive drug target.." "Gene A" should be "FYN"?

We thank the reviewer for catching this. Indeed, it is “FYN” and we changed it in line 154.

j. Line 149: Introduce Saracatinib, and make the reader aware that it actually mostly targets Src, and FYN with lower affinity.

We newly added text in lines 73 and 164 to indicate that saracatinib is an inhibitor against Src family kinases.

k. Line 469: "by the two sgRNA."→ "by the two sgRNAs".

Corrected

l. Throughout text/figures/figure legends, please check for consistency in the naming of cell lines, compounds, referring to figures etc. (E.g. MDA-MB-231/MDA MB 231/MDAMB-231 ; Fig. 1/Figure 1).

Corrected. Thank you for catching this error.

m. In Methods, frequently ug or uL are used instead of µg or µL

Corrected.

n. Legend Figure 5: Clarify what A, G, I, D, and P mean.

Corrected in line 685-686 to: “A: NVP-ADW742, G: gefitinib, I: imatinib, D: doxorubicin, P: Paclitaxel.”

o. Line 303: What is meant by: "The six variable nucleotides were added in reverse primer for multiplexing". Could you clarify this in the text?

We apologize for confusion the six nucleotides is index sequence for multiplexed run in NGS. The text in lines 373-374 is edited to: “The six nucleotides described as “NNNNNN” in reverse primer above represents unique index to identify biological replicates in multiplexed NGS run.”

**Reviewer #2 (Recommendations For The Authors):**
To enhance the robustness of the conclusions drawn from this study, certain concerns merit attention.Concerns:(1) Line 130 indicates that eight synergistic target gene combinations were validated. It would be helpful to clarify the criteria used to select these gene pairs and provide the rationale for studying these specific combinations of genes.

In fact, we had selected the gene pairs that we had the sgRNAs against available when we performed the experiments, so we did not have very good reason to explain our selections. Instead we added a brief discussion in lines 304-306 that further validations are required for the gene pairs not experimentally tested.

(2) According to Figure 2C, FYN was identified as crucial among the 30 gene pairs, and its upregulation in TNBC prompted further investigation. It would be informative to discuss the expression levels of TEK, FRK, and FGFR2 in TNBC and explain why these nodes were not studied. Is there existing evidence demonstrating the superiority of FYN over these other genes?

The similar concern was raised by reviewer #1. The expression levels of TEK, FRK and FGFR2 were relatively low in MDA-MB-231 and TNBCs in general, and we were concerned about the generalizability of these targets for treating TNBC. While the validation of these genes for possible synthetic lethality may lead to valuable insight, this may be beyond scope of this paper. This concern is newly discussed in result and discussion sections in lines 150-154.

(3) The screening process employed only one cell line, and validation was conducted with only one cell line (Figure 2A). Consider supplementing the findings with more convincing evidence from other breast cancer cell lines to strengthen the conclusions.

Although the CRISPR screens and primary validations were done with only one cell line, further validations with drug combinations were done in independent cancer cell lines such as Hs578T (figures S4E-J). Also, the possible association of FYN expression in drug tolerant cells were also demonstrated in lung cancer cells. We hope this satisfies the reviewer.

(4) The network analysis in Figure 2C lacks a description of the methodology used. It would be beneficial to provide a brief explanation of the methods employed for this analysis.

The network analysis was done manually with the size of each node proportional to the number of gene pairs. We newly added text in figure legend in line 638 to clarify this.

(5) The significance of gene A mentioned in line 142 is unclear. Please provide a clear explanation or context for the importance of this gene.

This is a mistake that were also pointed out by reviewer #1. The “gene A” should have been “FYN”. We corrected this in line 154.

6. In Figure 2J and Figure 2K, it would be more informative to measure the phosphorylation levels of FYN and SRC rather than just their baseline levels. Consider revising the figures accordingly.

We thank the reviewer for a careful comment. We newly provide supplementary figure S5A to show that phosphorylation level of FYN is increased, but this increase was proportional to the increase in FYN protein level, so the ratio of pFYN/FYN did not change significantly. We discussed this result in lines 187-190.

(7) Figure S4B lacks biological replicates, which could impact the reliability of the experimental results. Consider adding biological replicates to enhance the robustness of the findings.

This was also pointed out by reviewer #1. Instead of performing qPCR for all drugs, we focused on validating the decrease in FYN mRNA level for drug combinations that synergistically kill cancer cells. We were also aiming to identify direct mediator of FYN mRNA upregulation, so we focused on drug combination that involves inhibitor of epigenetic regulator that promotes transcription. To this end, we tested the impact of GSK-J4(KDM6 inhibitor) and pinometostat (DOT1L inhibitor) in combination with TKI in regulating FYN expression level. Notably, while GSK-J4 attenuated FYN mRNA accumulation by NVP-ADW742 treatment, pinometostat failed to do so (figure S5C). We newly described these results in lines 192-197 in results section.

(8) Line 186 indicates that KDM3 knockout was not tested in Figure S5A. It would be helpful to provide an explanation for this omission or consider including the data if available.

We thank the reviewer for pointing this out. The T7 endonuclease assay results for KDM3, KDM4 and PHF8 are added in figure S6B. All guide RNAs used in the study efficiently generated indel mutations.

(9) In line 206, KDM4A is introduced, but Figures 3J and 3M had already pointed to KDM4A. The authors did not analyze the ChIP results for other members of the KDM4 family at this point. Please address this inconsistency and provide a rationale for focusing on KDM4A. Additionally, in Figure 3M, consider adding peak labeling to the enriched portion for clarity.

We welcome the reviewer’s careful concern. KDM4 family enzymes perform catalytically identical reactions, and are thought to be redundant. Therefore, we judged that the most abundantly expression genes among KDM4 family should be the primary target to focus on. To this end, we analyzed the expression levels of KDM4 family genes in supplementary figure S6A. Indeed KDM4A expression was the highest among other KDM4 family genes. We discussed this in results section in lines 218-220.

(10) The author only indicated the relationship between the H3K9me3 level in the enhancer region and FYN expression. It would be valuable to verify the activity of the enhancers and investigate additional markers such as H3K27ac and H3K4me1. Consider discussing these aspects to provide a more comprehensive understanding.

Since we and others had shown that histone dementhylases are increased upon drug treatment, we focused on histone methylation marks which are associated with gene repression and whose removal by demethylases are associated with drug resistance. To this end, KDM6 demethylases removing H3K27me3 may serve as attractive alternative. In our newly added supplementary figure S6E, ADW742 treatment did not decrease H3K27me3 level in FYN promoter, indicating that H3K9me3 may be the dominant epigenetic change that modulates FYN expression upon drug treatment. This was briefly discussed in lines 233-235.

(11) In Figure 4A, the addition of the drug alone does not inhibit tumor growth. Please provide an explanation for this result and consider discussing potential reasons for the observed lack of inhibition.

The drug dose was adjusted carefully to minimize tumor shrinkage by single drug so that synergistic tumor shrinkage can be clearer.

(12) Line 208 indicates missing parentheses in the text describing Figure 4C. Please correct the text accordingly to ensure clarity.

Corrected. Thank you for catching this error.

(13) The figure legends for Figures 5E, F, G, and H contain errors. Please correct the figure legends to accurately describe the respective figures.

We thank the reviewer for catching this error. We have changed the figure legends in lines 691-697 to accurately describe the figures.

(14) It may be beneficial for the authors to divide the results section into several subsections and add headings to improve the overall understanding of the findings.

This is an excellent suggestion. We divided our results section into subsections and added headings in lines 80, 141, 181, 237 and 251 to help readers understand our findings.

(15) The authors should include the sgRNA sequences used for gene targeting, along with details of the target genes and negative/positive controls, in the Supplementary Materials to enhance reproducibility and transparency.

This is a critical point for improving reproducibility of our work. The sgRNA sequences used in the study are newly added in supplementary table S3.

(16) The resolution of the figures in the Supplementary Materials is too low, which may impede the authors' ability to interpret the data. Consider providing higher-resolution figures for better readability.

We had similar concern posed by reviewer #1, we provided higher resolution image for all main and supplementary figures.